# External and internal drivers behind the formation, vegetation succession, and carbon balance of subarctic fen margin.

Teemu Juselius-Rajamäki[1], Sanna Piilo[1], Susanna Salminen-Paatero[2], Emilia Tuomaala[1], Tarmo Virtanen[1], Atte Korhola[1], Anna Autio[3], Hannu Marttila[3], Pertti Ala-Aho[3], Annalea Lohila[4,5], Minna Väliranta[1]

[1]University of Helsinki, Ecosystem and Environmental Research Program
[2] University of Helsinki, Department of Chemistry, Radiochemistry
[3]University of Oulu, Water, Energy and Environmental Engineering Research Unit
[4]Finnish Meteorological Institute, Climate System Research Unit, P.O. Box 503, 00101 Helsinki, Finland
[5]Institute for Atmospheric and Earth System Research/Physics, University of Helsinki, P.O. Box 64, 00014, Finland

*Correspondence to*: Teemu Juselius-Rajamäki (teemu.juselius@helsinki.fi)

**Abstract.** Peatlands are the most dense terrestrial carbon storage and recent studies have shown that the northern peatlands have continued to expand to new areas to this day. However, depending on the vegetation and hydrological regime in the newly initiated areas, the climate forcing may vary. If these new areas develop as wet, fen-type peatlands with high methane emissions they would initially have a warming effect on the climate. On the other hand, if the development starts as dry bog-type peatlands, these new peatland areas would likely act as a strong carbon sink from early on. However, although some research has concentrated on the expansion of the new northern peatland areas, there remains a significant lack of studies on the successional development of the newly initiated peatland frontiers. In this research, we combined palaeoecological, remote sensing and hydrological modeling methods to study the expansion and successional pathway dynamics in a subarctic fen margin in Finnish Lapland and discussed possible implications for carbon balance of these margin peatland areas. Our results showed that the studied peatland margins had started to develop ca. 2000 years ago and have continued to expand thereafter, and this expansion has occurred in non-linear fashion. In addition, the wet fen-type vegetation persisted in the studied margin for majority of the development history and only the dryer conditions after the Little Ice Age instigated the fen-to-bog transition. However, a notable part of the fen margins in the Lompolovuoma and Lompolojänkkä basins has remained as a wet fen-types, and their persistence was likely caused by the hydrological conditions in the peatland and in the surrounding catchment. Our findings show a large variation in the peatland expansion and succession dynamics even within a single peatland basin. Although changes in climate conditions had initiated the fen-to-bog process in some margins, some had remained in the wet, fen stage showing resilience to allogenic forcings. Thus, when estimating the peatland carbon stocks, and predicting the future trajectories for peatland development, this heterogeneity should be taken into account to avoid errors caused by over-simplification of peatland lateral expansion dynamics.

## 1 Introduction

After peatland initiation through a primary peat formation, infilling (terrestrialization), or paludification, peatland area is increased by lateral expansion – the most important process of forming new peatland areas in modern climate

(Ruppel et al., 2013). In raised mires, these new peatland margin areas have been generally described as moist
minerotrophic fens and spruce swamps (Howie & Meerveld, 2011; Rydin & Jeglum, 2013), while in aapa mires
(patterned fens) the margins vary from dry ombrotrophic bogs to wetter lush swamps (Laitinen et al., 2005, 2007).
However, although the current vegetation in aapa mire margins has been described in a standard peatland literature,
there is an obvious lack of studies on the long-term successional development of these transitional ecotones between
peatlands and the surrounding mineral land. A recent study with main focus on aapa mire region of Finland showed
that the northern peatlands are still expanding (Juselius-Rajamäki et al., 2023), and whether these newly forming
peatland areas initiate and develop as moist fens or dryer bog-types can markedly affect the climate forcing of this
recent lateral expansion.

Lateral expansion process is driven by both allogenic and autogenic factors. For instance, forest fire or other
disturbance in areas adjacent to a peatland decreases the evapotranspiration and causes rising water table that enables
peatland expansion (Kuhry & Turunen, 2006). Similarly, waterlogging may be caused by autogenic development of
adjacent peatlands. As the peat accumulates vertically, the surface and groundwater flow pathways are directed
towards the margins of peat mound (Autio et al., 2023), creating suitable conditions for new peat formation (Korhola,
1996; Rydin & Jeglum, 2013). On the other hand, drainage ditches located in the mire margins can prevent natural
discharge to peatlands blocking the lateral expansion (Sallinen et al., 2019), while high-intensity fires can destroy peat
layers setting back the advance of peatland margins (Kuhry, 1994; Simard et al., 2007). Also, climate affects the lateral
expansion of peatlands, and for example, during warm and dry climate phase between 8000 – 5000 Before Present
(BP) expansion of peatlands slowed down, while wet and humid climate from 5000 to 3000 BP promoted lateral
peatland expansion (Korhola, 1994, 1995; Ruppel et al., 2013).

The development of vegetation communities in the newly initiated peatland margins vary according to the non-linear
successional trajectory and is driven, particularly by seasonal water availability, and consequently transportation of
essential ions (Goud et al., 2018). Depending on topography, surface flow may control the first appearance of
vegetation communities. Later groundwater seepage, point-scale or as wider seepage front, transports moisture and
dissolved elements for established plants. Compared to raised mires that have grown vertically above the surrounding
marginal areas, and often the entire landscape (Howie & Meerveld, 2011; Rydin & Jeglum, 2013), the secondary
peatland development pattern over the margins is more complex for aapa mires, because the shape of the peatland
varies from flat to concave (Seppä, 2002) and formation of new peatland habitats is dependent of water supplies from
snowmelt (Sallinen et al., 2023) and dilution of ion concentrations (pH-levels). Newly established habitat types may
range from ombrotrophic bog-types to minerotrophic swamps and fens (Foster & King, 1984; Laitinen et al., 2005,
2007; Mäkilä & Moisanen, 2007; Ruuhijärvi, 1983). However, mechanisms, such as surface water hydraulic forcing,
which create different types of margins, are currently poorly understood.

Differences in local hydrology mirrored in the current vegetation communities suggest opposite climatic feedback
mechanisms for the peatland centers and marginal areas. The overall climatic effect of peatlands is and has been
strongly controlled by the balance between sequestration of carbon dioxide ($CO_2$), and release of methane ($CH_4$)
(Frolking & Roulet, 2007). Methane is produced in anoxic conditions and released into the atmosphere via vegetation,
ebullition or by diffusion (Lai, 2009; Rydin & Jeglum, 2013). However, in areas where the acrotelm i.e., the oxic and
biologically active layer of the peat, is thick most of the methane is oxidized to carbon dioxide (Lai, 2009). Thus, in
the peatland margins where dry bog-type vegetation communities dominate, the climate forcing is most likely
negative, i.e, cooling impact on climate, due to the continuous uptake of CO2 and low CH4 emissions. However, if
the water table depth becomes too deep, accelerated decomposition can turn these locations to carbon sources due to
increased CO2 emissions that negate the decrease in CH4 emissions (Evans et al., 2021). On the other hand, in wet
fen-type margins high methane emissions have an opposite effect on short timescales, further amplified by graminoid
vegetation communities (Bubier et al., 1993; Juutinen et al., 2013; Kou et al., 2022; Ward et al., 2013).

Often, the interest of (palaeo)peatland researchers has been in the deepest and oldest part of a peatland while the
development of peatland margins, i.e., young areas, has attracted less consideration (Korhola et al., 2010; Ruppel et
al., 2013). Only recently the focus has turned to peatland margins and peat profile sampling has been extended to the
peatland-upland ecotones (Juselius-Rajamäki et al., 2023; Lacourse et al., 2019; Le Stum-Boivin et al., 2019;
Mathijssen et al., 2014, 2016, 2017; Peregon et al., 2009; Schaffhauser et al., 2017). Even these studies have focused
more on the expansion dynamics of the peatlands, while the vegetation succession of the marginal areas has deserved
lesser consideration. As the past vegetation communities can be used to ascertain climate feedback, the knowledge of
vegetation succession in peatland margins can be used to better understand how lateral expansion has affected the past
climates and helps us to predict the effects of lateral expansion for future climate change.

Here, we studied the expansion and successions pathways of peatland margins in a subarctic fen, Lompolovuoma,
located in Finnish Lapland using a novel approach combining palaeoecological, remote sensing and hydrological
modeling methods. The study was conducted across three transects, from the edges towards the centre of the peatland,
with each transect having three peat profiles. The vegetation succession was studied by a high-resolution plant
macrofossil analysis, and AMS ($^{14}$C) radiocarbon dating was carried out to date the basal peat layers and the major
plant compositional shifts, respectively. To have a wider understanding of development and diversity of plant
communities in aapa mire margins, we used additional comparable peat profile data from three peatlands from northern
Finland as well as detailed remote sensing-based vegetation and land-cover classification (Räsänen et al., 2021) from
Lompolovuoma fen margins. Finally, water table depth and groundwater-surface water interaction fluxes derived from
a fully integrated hydrological model (Autio et al., 2023) were used to demonstrate the connections between altered
drier and wetter climatic conditions and peatland vegetation succession. The results of our study give an insight into
aapa mire margin succession patterns, their relation to hydrology, and a basic understanding of the peatland climate
feedback and carbon balance related to peatland lateral expansion in subarctic areas.

## 2 Methods and materials

### 2.1 Study sites

The Lompolovuoma study site is a subarctic fen located in the municipality of Muonio in Finnish Lapland (67° 59' 42" N, 24° 12' E, Fig. 1a). The site belongs to the northern aapa mire zone with more continental climate, shorter growing season, and more profound frost effects than on the aapa mires located further south (Ruuhijärvi, 1983). The mean annual temperature in the study site is 0.4 °C (2003-2019) and the mean annual precipitation 647 mm (2008-2019) (Marttila et al., 2021).

We studied the margins of a sub-basin in a larger fen complex that comprises of several elongated, north-south aligned fen areas. The vegetation in the central areas of the study site is dominated by typical wet fen taxa, such as various *Carex* species and flark *Sphagnum* species. Strings are mainly occurring in the southern parts of the basin. A stream runs across the peatland basin from south to north towards Lake Pallasjärvi.

Vegetation communities in the studied peatland margins resemble raised pine bog habitats in the south with low hummocks and narrow lawn areas (Laine et al., 2018). The ground layer consists of *Sphagnum fuscum* and *Sphagnum angustifolium,* and Cladonia sp. lichens also occurred. In the field layer *Eriophorum vaginatum*, *Rubus chamaemorus* and various dwarf shrubs such as *Empetrum sp., Andromeda polifolia and Vaccinium vitis-idaea* was found. In addition, stunted *Pinus sylvestris* grow on the hummocks.

To expand our understanding of vegetation succession in aapa mire margins, we used three additional short profiles collected from aapa mires elsewhere in Finland: Syysjärvi, Salamajärvi and Patvinsuo (Fig. 1a). These profiles enabled comparison between different local and geographic settings across Finland. For a full description of the study sites, field sampling, and laboratory analysis for supplementary sites, see Juselius-Rajamäki et al. (2023).

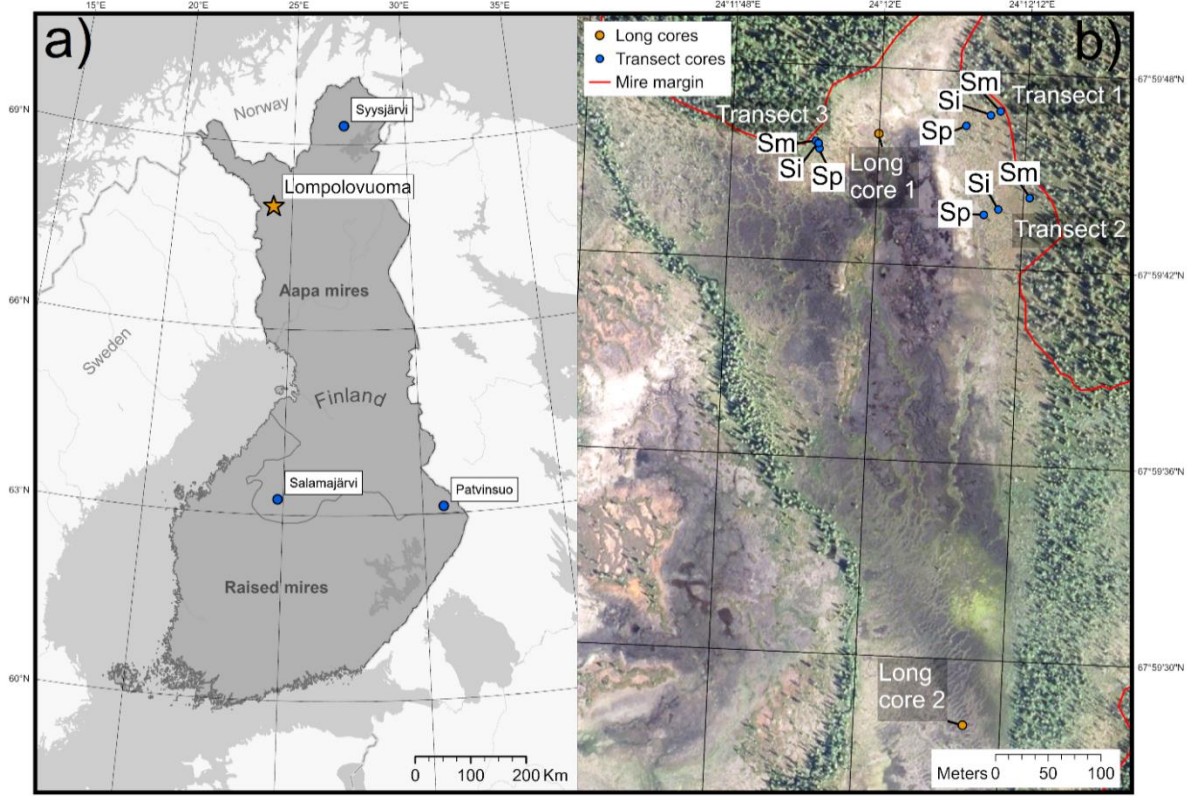

129

Figure 1. a) Location of the main study site Lompolovuoma marked with an orange star while locations of the comparison
sites are marked with blue circles. Borderline separating aapa and raised mire complex areas in Finland is presented. 1. b)
The study location within Lompolovuoma basin shows the study transect samples in blue circles, and the long cores in
orange circles. For the transect samples the sample code indicates the sample location within the transect: and Sm for the
sample closest to the mire margin, Si for the sample in the middle of the transect and Sp for the sample closest to the
peatland center. Mire margins are shown with a red line. Aerial image from National Land Survey of Finland, taken in
2023. This figure contains data from National Land Survey of Finland NLS Aerial photographs database.

## 2.2 Field sampling

The field sampling for the study was conducted during the summer and the autumn of 2022. To study the lateral
expansion dynamics, we sampled a total of three transects coded as T followed by transect number with three peat
core samples coded as S followed by core location indicator: p for sample located closest to the peatland centre, i for
the intermediate sample, and m for the sample located closest to the margin. Each transect ran from the edge of the
peatland towards the centre. The transects were placed on the "bog-type" margin with variable width, which led
variable lengths in our transects. All profiles were collected from lawn-microform with similar types of vegetation.
The first sampling location for each transect was at the extreme end of the "bog-type" margin next to the central fen
area. The other samples were collected along the transect from locations fulfilling the above criteria, and in such
manner that entire length of "bog-type" -margin at the transect location was covered. We established two transects
from the east edge and one transect from the north edge of the fen sub-basin (Fig. 1b). The peat cores were taken with
a box corer ($7 \times 4 \times 65$ cm) down to mineral subsoil. To reconstruct Holocene peatland initiation, in addition to the
peat cores sampled from the mire margin, four long cores were collected from two different locations of the central
part of the study basin: two of the long cores, a and b, were respectively located close to each other and are representing

replicates (Fig. 1b). These samples were collected using a Russian peat corer (3 × 50 cm). The profiles were described
and classified in the field, and the length of the profile was measured. The location of each sampling point was recorded
using Trimble R8 GPS device with ± 0.05 m accuracy and the distance between each transect sampling point was
measured using a tape measurer. After sampling, the peat cores were carefully wrapped in plastic to avoid any
contamination and transported to University of Helsinki premises. The samples were stored in a cold room prior to
further analysis.
**2.3 Laboratory analysis**
The short profiles were cut into 1-cm subsamples and from these subsamples, dry bulk density (BD, g/cm$^3$) and
sediment organic matter (OM) based on the loss on ignition method (LOI) were determined (Heiri et al., 2001). We
used LOI values to differentiate between the mineral subsoil and the peat. We defined peat initiation depth based on
the first layer where LOI ≥ 70 % (Korhola, 1994). In addition, we analyzed the C/N-content as following: 4 cm interval
(transect 1 and 3) and 5 cm interval (transect 2) using LECO TruSpec micro-Elemental Determinator. For the long
profiles, contact layer between limnotelmatic *Equisetum* peat and fen peat, without visible Equisetum remains, was
first determined in the field and then confirmed using a stereomicroscope.

To reconstruct past changes in vegetation, plant sub-macrofossil analysis for each short peat profile was conducted at
4 cm intervals and when prominent changes occurred the interval was increased to every second cm. The percentage
proportion of each peat forming vegetation type of a total sample volume (100 %) was analyzed from 5 cm$^3$ peat
samples that were gently rinsed under running water in a 100 µm sieve. The residue was analyzed under a
stereomicroscope following Väliranta et al. (2007) and Mauquoy et al. (2014). For example, seeds and leaves were
counted in exact numbers and the percentage of unidentified organic material (UOM) estimated for highly decomposed
organic remains that had lost their microscopical characteristics. A compound light microscope was used for higher
taxonomic level identification. Software Tilia (Grimm, 1991) and C2 (Juggins, 2007) were used to create diagrams.

To study the lateral expansion and succession dynamics of the fen margins, we applied AMS radiocarbon ($^{14}$C)
determinations to date the basal peat of each short profile and the depths corresponding to the major regime shifts in
vegetation e.g., first occurrence of the *Sphagnum* mosses overlying sedge-dominated peat and the shift to *Sphagnum*
dominance. For the long profiles we dated the shift from limnotelmatic *Equisetum* peat to fen peat to gain
understanding of long-term development of the Lompolovuoma fen. Terrestrial plant remains and/or charcoal were
prioritized for $^{14}$C analyses over bulk peat samples (Quik et al., 2022). However, in three cases regarding the short
cores, the peat was highly decomposed and bulk peat had to be used (Table 1). In addition, bulk peat was used as
material for the AMS dating of the long cores. Rootlets were carefully removed from the bulk peat samples. Samples
were dated in Poznan Radiocarbon laboratory (Poznan, Poland). We calibrated $^{14}$C BP ages against the INTCAL 20
NH calibration curve (Reimer et al., 2020) and modern dates (pMC % modern carbon) by using the Bomb21 NH1
calibration curve (Hua et al., 2022). Finally, calibrated ages were rounded to the nearest 5 years.

For the comparison profiles, radiocarbon dating results were acquired from Juselius-Rajamäki et al. (2023). In
addition, radiolead ($^{210}$Pb) dating was performed for the comparison profiles at the Department of Chemistry,
University of Helsinki. The separation method used for $^{210}$Po was a combination of several previously published
methods (Ali et al., 2008; Flynn, 1968; Kauranen & Miettinen, 1966; Sanderson, 2016). Dried peat samples were
digested with concentrated acids $HNO_3$ and $HCl$. $^{209}$Po tracer spike was added to the samples at the beginning of the
analysis to monitor the yield loss. After digestion, the samples were evaporated to dryness, dissolved into a dilute HCl
solution, filtered, and transferred into deposition vessels made from PTFE. Ascorbic acid was added to reduce
interfering impurities, e.g. Fe, in the samples. $^{210}$Po was deposited spontaneously onto a silver disc in the deposition
vessel using a heated water bath (65-75 °C) with constant stirring for 2.5-3 hours. The activity concentration of $^{210}$Po
was measured from the silver disc with a PIPS (passivated implanted planar silicon) detector. The activity
concentration of $^{210}$Pb in the samples was obtained via equilibrium of $^{210}$Po and $^{210}$Pb in the samples.

**2.4 Age-depth models**

Age-depth models with $^{14}$C ages were done using Bacon package ver. 3.2.0 (Blaauw & Christen, 2011) in R software
ver. 4.3.1 (R Core Team, 2023). We assumed different peat accumulation rates for different vegetation community
stages, and these were acquired from the literature representing similar vegetation communities and geographic
locations (Granlund et al., 2022; Mäkilä et al., 2001; Mathijssen et al., 2014; Rydin & Jeglum, 2013; Zhang et al.,
2020). We used these accumulation rates as a prior value for the age-depth models for corresponding vegetation
community stages. After the initial model run, if the model fit was not satisfied (Blaauw & Christen, 2011), the prior
values were altered to ensure the model fit. Boundaries were set for the profiles based on vegetation community shifts,
and different accumulation rates were calculated for different plant communities. For the profiles with both $^{14}$C and
$^{210}$Pb ages, e.g., (SyJ T1Sm, SJ T3Sm, and PS T1Sm), we used Plum package ver. 0.4.0 (Aquino-López et al., 2018)
in R software ver. 4.3.1 (R Core Team, 2023). For the comparison peat profiles, the same prior accumulation rates
were used as for the Lompolovuoma study site. Again, to accommodate for the individual peat profile characteristics,
the rates were modified to ensure age-depth model fit. The individual age-depth models containing the accumulation
rates and used boundaries are presented in supplementary data (figures 1-12).

**2.5 Lateral expansion rates**

Lateral expansion rates (cm/year) were calculated between adjacent peat sections in each transect. The rates were
calculated by dividing the horizontal distance between adjacent dated profiles (cm) with the difference of the basal
ages, respectively (years). Mean calibrated ages from the age-depth model were used.

**2.6 Current vegetation community analysis**

We used field and remote sensing-based land cover type data from Räsänen et al. (2021), where the methodology is
described in detail, to estimate the proportion of vegetation communities in the peatland margins. Land cover
classification was based on field verification data collected in summer 2019, and multisource remote sensed data.
Classification was conducted in two steps: first 4 channel 0.5 m pixel sized aerial image from summer 2018 was
segmented, and then for these segments (mean size 50 m2) values were calculated from several Lidar, Planetscope
and Sentinel images from years 2018 and 2019, and these were classified using random forest classification. Final
land cover product had 16 classes, and the overall classification accuracy was 76%. Here, we used a simplified
classification based on ombrotrophic – minerotrophic gradient to describe habitat conditions and related vegetation
community. In addition, tree-covered fens were separated from open fens. Applied vegetation communities are: "bog"-
type (referring to dry conditions), "fen"-type (referring to wet conditions), and tree-covered fens (referring to forested
peatland) and these enable comparison with the remote sensing data. These were combined from the land cover type
classes with similar ecological characteristics: dwarf shrub pine bogs and dwarf shrub bogs as the bogs, tall sedge fens
and flarks as the fens and paludified spruce, birch, and mixed forests as the tree-covered fens. We delineated our study
basin Lompolovuoma and adjacent Lompolojänkkä basin based on the land cover dataset in ArcGis Pro ver. 3.1.0
(ESRI, 2023) and calculated the proportion of each land cover type for the whole peatland area and for the peatland
margins. For the peatland margins, we chose a 25-meter distance from the peatland forest border to represent the
marginal peatland area. This distance prevented any overlap of the marginal areas even in the narrowest parts of the
peatland and allowed non-biased analysis of the marginal peatland types irrelevant to the topography or vegetation on
site.
**2.7 Hydrological analyses**
To study the hydrological drivers behind the development of divergent peatland types at the fen margins detected in
our vegetation coverage analysis, we used the fully integrated physically based-hydrogeological model
HydroGeoSphere (Aquanty, 2015). The model allows explicit simulation of water exchange between groundwater
and surface water and can be parameterized using physical properties of peat and mineral soils. The high spatial
resolution of the model makes it suitable to estimate water fluxes at the scale of vegetation inventories and remote
sensing data. This model has been previously implemented for the Pallaslompolo catchment, and the full methodology
for this hydrogeological model is described in Autio et al. (2023). Due to the original study boundaries, this model
only covers Lompolojänkkä sub-basin. In this study, we (1) investigated the resulting hydrological conditions in terms
of groundwater-surface water exchange flux and (2) compared the impact of the current (baseline) and the drier climate
in terms of water table elevation (Helama et al., 2017) .

In (1), we investigated the prevailing groundwater-surface water exchange fluxes of the transient model run averaged
over the summer of 2017 within each peatland type. For (2), we studied the effect of drier climate conditions by
comparing the outputs of the steady-state simulations for the current climate with the effective rainfall $P_{eff}$ equal to
385 mm (average for 2016-2018) and the drier climate of $P_{eff}$ equal to 250 mm. The value of 250 mm is within the
measured range that varied between 170 mm and 574 mm in 2008-2018 but represents a significantly lower value
than the measured long-term mean of 358 mm for the years (2008-2018). Due to the variable density of the model
computing mesh, the model output was first plotted in the postprocessing visualisation software Tecplot 360 EX 2022
R2, which accommodates value interpolation over element size. The variables were divided into separate bins
according to magnitude, hereafter referred to as contour groups showing spatial variation in model output. The
resulting raster image was imported to GIS mapping software (ESRI, 2023), georeferenced and clipped according to
the defined peatland margins for each peatland type. The areas of each contour group were then calculated respectively
for each peatland type.

## 3 Results

### 3.1 Peat initiation and spatial development of the peatland margins

In transect 1, the oldest basal date ca. 2230 cal BP was dated from the peat profile closest to the mire center (T1Sp)
(Table 1). For the intermediate profile (T1Si) the basal age was ca. 1185 cal BP and for the profile next to the forest
(T1Sm) basal age was ca. 990 cal BP. In transect 2, the oldest basal age found in the intermediate profile (T2Si) was
1930 cal BP while younger basal ages of 1025 cal BP and 390 cal BP were found for the T3Sp and T3Sm, respectively
(Table 1) Oldest basal age in transect 3 was 1390 cal BP in the intermediate sample T2Si (Table 1) while the basal
age in the sample closest to the mire center (T3Sp) was 1225 cal BP and in the peatland margin (T3Sm) 765 cal BP.

Long core (LC) dating results suggest that a shift from limnotelmatic peat to fen peat occurred ca. 6300 cal BP at
earliest and around 4000 cal BP at latest (Table 1). This change occurred earlier in the northern part of the sub-basin
(LC1a ca. 6290 cal BP and LC1b ca. 6360 cal BP). In the southern part, this shift occurred ca. 4865 cal BP for LC2a
and ca. 4365 cal BP for LC2b.
**Table 1. Peat profile description. Coring location describes the location of the sampling across the transects with "Margin"**
**being located closest to the mire-forest boundary, and "Peatland" closest to the mire center. Sample type describes the**
**location within the profile with "Basal" representing the contact layer between peat and mineral subsoil, "Sphagnum**
**occurrence" indicating the first occurrence of Sphagnum mosses and "Sphagnum dominance" the first layer with clear**
**Sphagnum-dominance. Sample description indicates material used in $^{14}C$ analyses. Age (cal BP) with 95 % confidence**
**interval show calibrated median age with 95.4 % confidence intervals.**

| Laboratory code | Core code | Sample location | Sample type | Depth (cm) | Dated material | $^{14}C$ Age (BP) | ± | pMC | ±2 | Age (cal BP) with 95.4 % confidence interval |
|---|---|---|---|---|---|---|---|---|---|---|
| Poz-162912 | T1Sm | Margin | *Sphagnum* occurrence | 7-8 | *Sphagnum* and feather moss leaves and stems | | | 103.46 | 0.33 | -60 (-5 – -65) |
| Poz-162911 | T1Sm | Margin | Basal | 19-20 | Bulk | 1085 | 30 | | | 990 (1060 – 930) |
| Poz-162914 | T1Si | Intermediate | *Sphagnum* dominance | 27-28 | *Sphagnum* moss leaves and stems, woody | 315 | 30 | | | 390 (460 – 305) |
| Poz-162913 | T1Si | Intermediate | Basal | 30-31 | Woody | 1250 | 50 | | | 1185 (1285 – 1065) |
| Poz-165854 | T1Sp | Peatland | *Sphagnum* dominance | 21-22 | *Sphagnum* moss leaves and stems | | | 121.63 | 0.35 | -35 (-5 – -35) |
| Poz-162924 | T1Sp | Peatland | *Sphagnum* occurrence | 36-37 | *Sphagnum* moss leaves and stems, woody | 845 | 30 | | | 740 (790 – 685) |
| Poz-162925 | T1Sp | Peatland | Basal | 40-41 | Woody, charred wood | 2210 | 30 | | | 2230 (2325 – 2125) |
| Poz-162917 | T2Sm | Margin | *Sphagnum* dominance | 25-26 | *Sphagnum* moss | 85 | 30 | | | 115 (260 – 25) |

| Lab code | Profile | Location | Type | Depth (cm) | Material | | | | | |
|---|---|---|---|---|---|---|---|---|---|---|
| | | | | | leaves and stems, woody | | | | | |
| Poz-162916 | T2Sm | Margin | Basal | 29-30 | Shrub leaves, woody, bulk | 320 | 35 | | | 390 (470 – 305) |
| Poz-165855 | T2Si | Intermediate | *Sphagnum* dominance | 32-33 | *Sphagnum* moss leaves and stems | 75 | 30 | | | 115 (260 – 30) |
| Poz-162920 | T2Si | Intermediate | *Sphagnum* occurrence | 41-42 | *Sphagnum* and feather moss leaves and stems | 570 | 70 | | | 590 (665 – 505) |
| Poz-162918 | T2Si | Intermediate | Basal | 48-49 | Woody | 1995 | 30 | | | 1930 (1995 – 1835) |
| Poz-162922 | T2Sp | Peatland | *Sphagnum* dominance | 35-36 | *Sphagnum* and feather moss leaves and stems | 150 | 30 | | | 145 (285 – 50...) |
| Poz-162921 | T2Sp | Peatland | Basal | 46-47 | Woody | 1140 | 30 | | | 1025 (1175 – 960) |
| Poz-165856 | T3Sm | Margin | *Sphagnum* dominance | 8-9 | *Sphagnum* moss leaves and stems, woody | | | 107.25 | 0.33 | -55 (-5 – -60) |
| Poz-162880 | T3Sm | Margin | Basal and *Sphagnum* occurrence | 19-20 | Woody, charred wood | 870 | 30 | | | 765 (905 – 690) |
| Poz-165857 | T3Si | Intermediate | *Sphagnum* dominance | 14-15 | Bulk with majority (>95%) of Sphagnum, woody | | | 109.35 | 0.34 | -50 (-5 – -55) |
| Poz-162619 | T3Si | Intermediate | Basal and *Sphagnum* occurrence | 32-33 | Woody | 1520 | 30 | | | 1390 (1515 – 1315) |
| Poz-165859 | T3Sp | Peatland | *Sphagnum* dominance | 13-14 | Bulk with majority (>95 %) of Sphagnum | | | 135.14 | 0.35 | -25 (-25 – -30) |
| Poz-162923 | T3Sp | Peatland | *Sphagnum* occurrence | 21-22 | Woody | 105 | 30 | | | 110 (270 – 15) |
| Poz-162882 | T3Sp | Peatland | Basal | 40-41 | Woody | 1290 | 30 | | | 1225 (1290 – 1175) |
| Poz-165876 | LC1a | Fen lawn | Fen peat | 127-129 | Bulk peat with roots removed | 5490 | 40 | | | 6290 (6395 – 6200) |
| Poz-165959 | LC1b | Fen lawn | Fen peat | 123-124 | Bulk peat with roots removed | 5595 | 35 | | | 6360 (6445 – 6300) |
| Poz-165085 | LC2a | Fen lawn | Fen peat | 180-181 | Bulk peat with roots removed | 4305 | 35 | | | 4865 (4965 – 4830) |
| Poz-165086 | LC2b | Fen lawn | Fen peat | 189-191 | Bulk peat with roots removed | 3930 | 35 | | | 4365 (4515 – 4245) |


## 3.2 Peat properties

A shift from mineral layer to organic layer was sharp in all profiles and in the upper parts of the profiles, the loss on ignition (LOI, %) values varied only slightly (Fig. 3, 4, 5). In transect 3, the mineral material has intruded into the peat at depths of 14 cm (T3Sp) and 23 cm (T3Si). Compared to LOI (%), more fluctuations were visible in bulk density (BD, g/cm$^3$) values (Fig. 3, 4, 5). Above the sharp mineral subsoil – peat contact, the BD decreased towards the surface with the lowest values found at the top of peat profiles. A stepwise decrease in BD occurred in peat profiles T1Sp, T2Si, T2Sp and T3Sm while a gradual decrease in BD values was observed in other profiles. Carbon content (%) above the mineral subsoil contact varied only slightly along the peat profiles (Fig. 3, 4, 5) and highest nitrogen contents along the peat profiles were found in the layers closest to the mineral subsoil and the surface (Fig. 3, 4, 5).

**3.3 Fossil plant communities and succession of the peatland margins**

Three main vegetation stages were identified in the Lompolovuoma margin peat profiles (Fig 2a-c, Fig 3-5). The first and oldest stage consisted of the remnants of cyperaceous and ericaceous vegetation (C-E), but lacked brown mosses usually associated with calcareous fens. This phase was characterized by a high proportion of unidentified organic matter (UOM), indicating a high level of humidification. The second stage contained remains of mixed *Cyperaceae-Ericaceae-Sphagnum* (C-E-S) vegetation. The transition from stage 1 to 2 occurred gradually in some peat profiles while sometimes shift was abrupt. In this transition *Sphagnum* sect. *Acutifolia* started to replace cyperaceous vegetation. In the transects 1 and 3 the high level of decomposition prevented species-level identification of *Sphagnum* mosses in early C-E-S stage. However, in transect 2, the C-E stage was directly overlain *by Sphagnum fuscum*. In the final *Sphagnum-Ericaceae* stage (S-E), the plant community is dominated by *Sphagnum* mosses, and the cyperaceous vegetation is nearly or completely missing. *Sphagnum* species consists of *Sphagnum fuscum*, *S. capillifolium, S. russowii*, and *S. angustifolium*. A varying amount (%) of ericaceous vegetation is usually mixed with the sphagna. Varying amounts of forest bryophytes, such as *Pleurozium schreberi* is also detected through the peat layers. In addition, in the marginal profiles of transect 1, the mire vegetation was replaced twice by forest vegetation, and similar replacement occurred once in the margin of transect 2. Macrofossil data is presented in Fig. A2, A3, and A4.

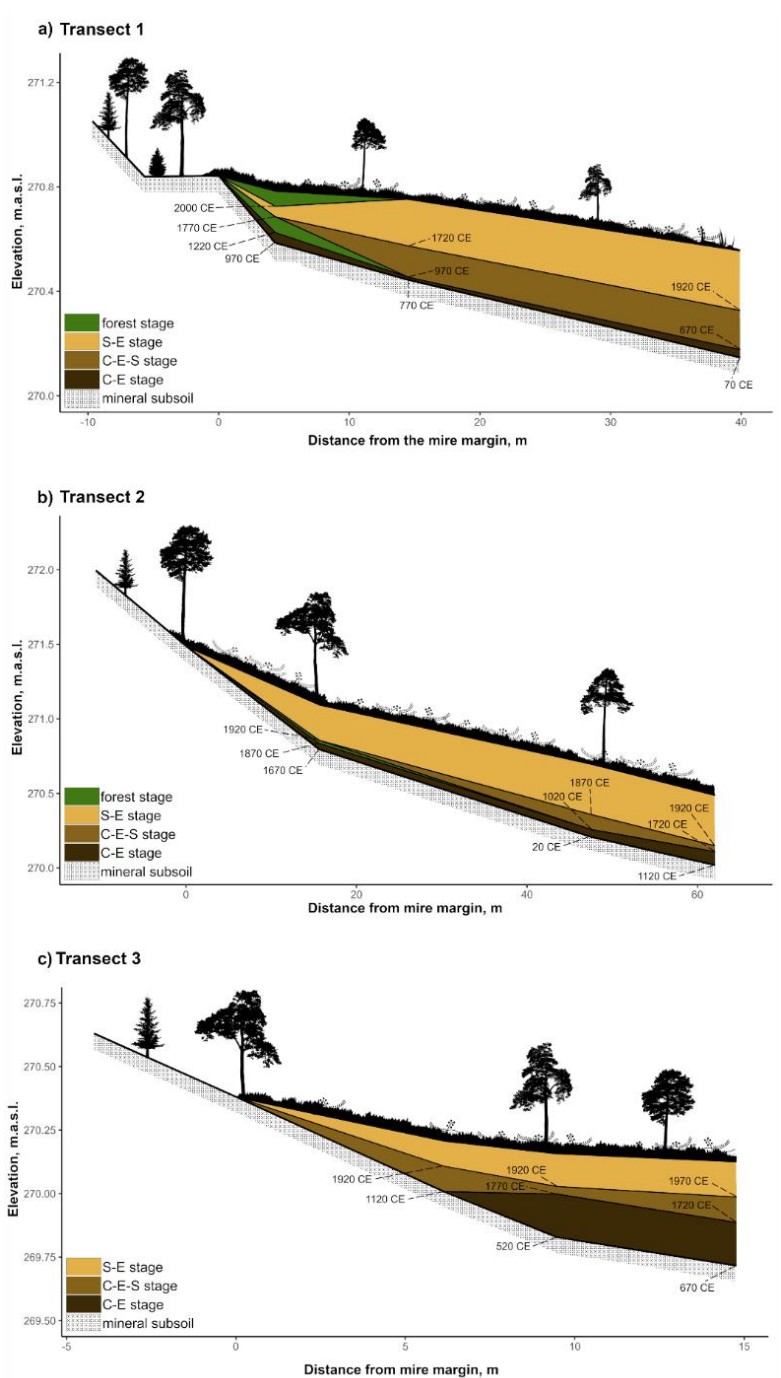

**Figure 2a-c. Transect profiles. The figures show the vegetation community stages: C-E (*Cyperaceae-Ericaceae*), C-E-S (*Cyperaceae-Ericaceae-Sphagnum*), and S-E (*SphagnumEricaceae*) as well as the forest community stages in the margins of T1 and T2. In addition, the onset of each stage at the location peat profiles is shown with ages (CE = Common Era) derived from the age-depth model. The ratios between the x and y axes varies between the illustrations. The vegetation is presented to give a rough impression of real-life conditions in the study transects locations and is not in the true scale.**

At the onset of peat development in the mire margins, the C-E vegetation community dominated (Fig. 2a-c, Fig. 3-5). In transects 1 and 2 this layer was thin, only up to 4 cm in transect 1 and from 5 cm to 9 cm in transect 2. In the transect 3 the C-E layer was markedly thicker, 16 cm in T3Si and 18 cm in T3Sp. Duration of the C-E stage was highly

variable: in transect 1 the C-E stage lasted between ca. 250 (T1Sm) and 600 years (T1Sp). In transect 2 C-E stage
lasted between ca. 200 (T2Sm) and 1000 (T2Si) years. In transect 3 the C-E stage was missing from the profile closest
to the mire margin (T3Sm), and *Sphagnum* mosses established directly on top of the mineral subsoil. The duration of
the C-E stage in T3Si was ca. 1250 years and in T3Sp ca. 1050 years.

The C-E stage ended asynchronously across Lompolovuoma mire margin and in most of the cases the C-E stage was
followed by the mixed C-E-S stage where sphagna started to colonize the margins. The establishment of sphagna
marking the start of the C-E-S occurred between ca. 670 and 970 CE in transect 1, between ca. 1020 and 1720 CE in
transect 2, and between ca. 1720 and 1770 CE in transect 3. No C-E-S stage was detected in samples T1Sm and T2Sm.
Instead, the vegetation shifted towards a mix of ericaceous vegetation, *Pleurozium schreberi* and *Dicranum* sp.
Suggesting turn to dryer conditions. In T3Sm, the C-E-S stage occurred directly over the mineral subsoil.

On contrary to asynchronous shift from C-E stage to C-E-S stage, the change to ombrotrophic vegetation community
(S-E) with high proportion of sphagna appeared nearly simultaneously across all studied margins. This stage started
between ca. 1870 and 1970 in all peat sections in transects 2 and 3 and similarly also in T1Sp. Only in T1Sm (1770
CE) and T1Si (1720 CE) the shift to S-E vegetation community stage occurred earlier. Currently S-E vegetation type
is predominant across the transects.

A comparable successional pathway as in Lompolovuoma was detected from Syysjärvi study site in eastern Lapland
(Fig. A1). A 1-cm thick ericaceous vegetation layer overlaid mineral soil, and this community was shortly replaced
by a 2-cm thick C-E layer similar to the results found in Lompolovuoma. These stages lasted only ca. 15 years,
respectively, after which C-E-S stage with some sphagna took over ca. 1970 CE. Above 3-cm thick C-E-S stage, the
S-E stage mostly comprised by *Sphagnum capillifolium* that took over in ca. 1980 and has persisted ever since.

Different successional pathways were found from Salamajärvi and Patvinsuo peatland sites (Fig. A1). In Salamajärvi,
there was no evidence of cyperaceous vegetation. Rather, the peat layers comprising of ericaceous vegetation with a
small amount of *Sphagnum* mosses initiated directly on mineral subsoil in ca. 1830 CE in the margin of the Salamajärvi
peatland. Afterwards, proportion of sphagna gradually started to increase and *Sphagnum* mosses became dominant ca.
1950 CE. Currently, *Sphagnum capillifolium* is the dominating moss species.

When peat formation started in Patvinsuo margin (Fig. A1) ca. 1850 CE, the initial vegetation consisted of C-E-S
vegetation. At first, proportion of *Sphagnum* mosses started to increase, and ca. 1915 CE those were the dominant
taxa. However, between ca. 1915 and 1950 CE *Sphagnum* mosses together with remains of *Cyperaceae* nearly
disappeared and mostly ericaceous vegetation remained and supplemented by the presence of *Cenococcum* sclerotia
that suggest dry mire margin conditions. However, towards present, the amount of *Sphagnum* mosses again increased
and currently they form most of the coring site vegetation, with *Sphagnum russowii* being the most common species.

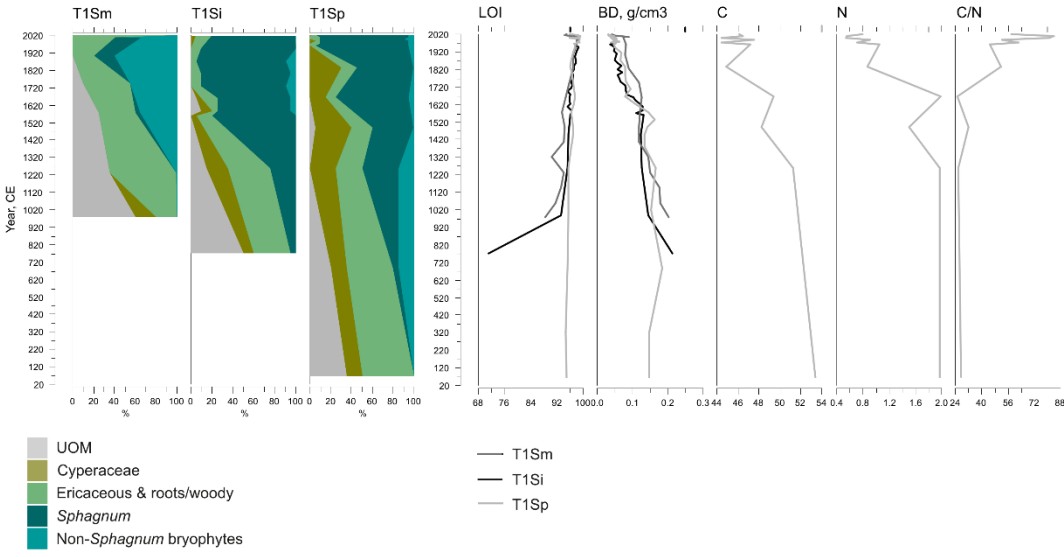

**Figure 3. Fossil plant records (left) including undetected organic matter (UOM) and loss on ignition (LOI), bulk density (BD) and carbon and nitrogen contents and C/N ratio (right) for transect 1. Proportion of vegetation type and LOI in percentages (%), unit for bulk density is g/cm³. Carbon content (%), nitrogen content (%), and C:N ratio is available for profile T1Sp only.**

**Transect 2**

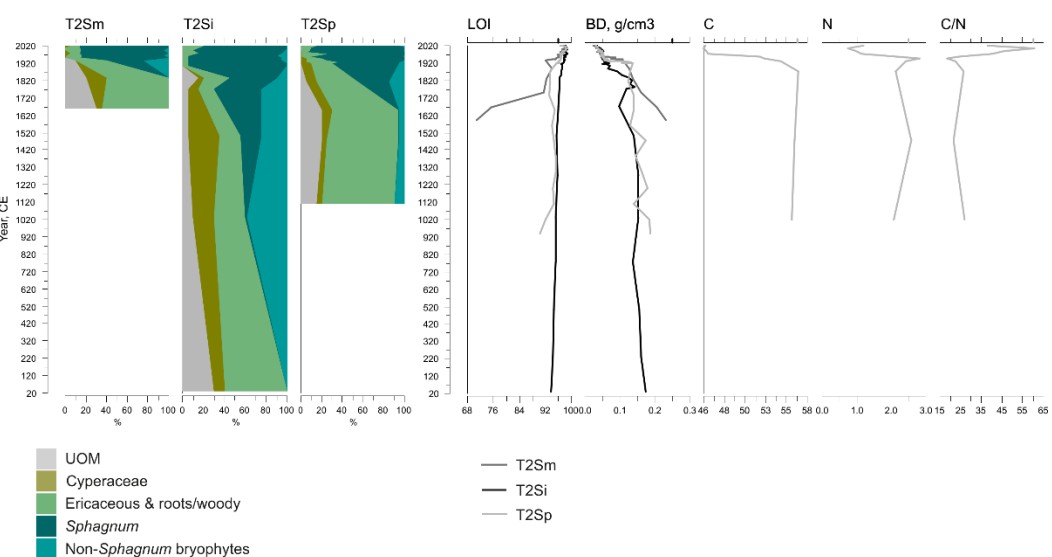

**Figure 4. Fossil plant records (left) including undetected organic matter (UOM) and loss on ignition (LOI), bulk density (BD) and carbon and nitrogen contents and C/N ratio (right) for transect 2. Proportion of vegetation type and LOI in percentages (%), unit for bulk density is g/cm³. Carbon content (%), nitrogen content (%), and C:N ratio is available for profile T2Sp only.**

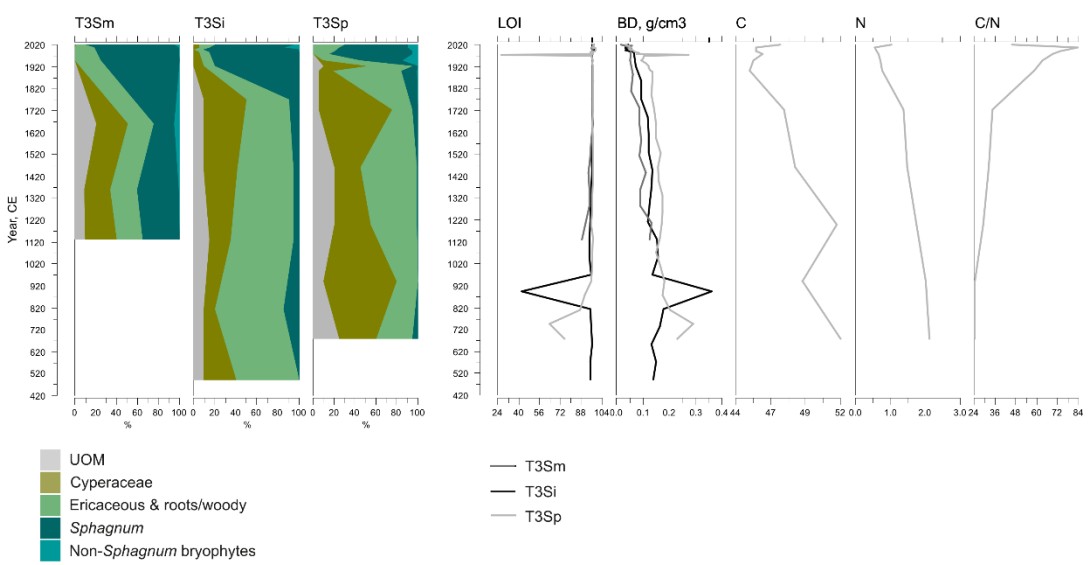

361

**Figure 5. Fossil plant records (left) including undetected organic matter (UOM) and loss on ignition (LOI), bulk density (BD) and carbon and nitrogen contents and C/N ratio (right) for transect 3. Proportion of vegetation type and LOI in percentages (%), unit for bulk density is g/cm3. Carbon content (%), nitrogen content (%), and C:N ratio is available for profile T3Sp only.**

## 3.4 Lateral expansion rates and vertical peat increment

The average rate of lateral expansion between dated peat profiles varied from 0.53 cm/year (T3Si to T3Sm) to 5.23 cm/year (T1Si to T1Sm). The median lateral expansion rate for all transects was 2.25 cm/year with interquartile range of 1.72 – 2.90 cm/year.

## 3.5 Vegetation community cover analysis

The total area of Lompolovuoma and Lompolojänkkä peatland basin is 141.2 ha, of which 34 % is classified as a bog-type, 46 % as fen-type, and 21 % as tree-covered fen (Table 2). The area 25 meters from the peatland border is in total 43.9 ha and covers 31 % of the total peatland area. In these marginal areas, bog type constituted 44 %, fen-type 23 %, and tree-covered fens 33 % of the mire margin area (Fig. 6, Table 2). In Lompolovuoma basin, where our study transects were located, the coverage of bog-type in the peatland margin is 54 % while in adjacent peatland basin, Lompolojänkkä, bog-type is covering smaller area, 35 %. On the contrary, higher coverage of fen-type is found in the margins of the Lompolojänkkä (26 %) than in Lompolovuoma (20 %). Similarly, larger areas were covered by tree-covered fens in Lompolojänkkä (39 %) than in Lompolovuoma (26 %).

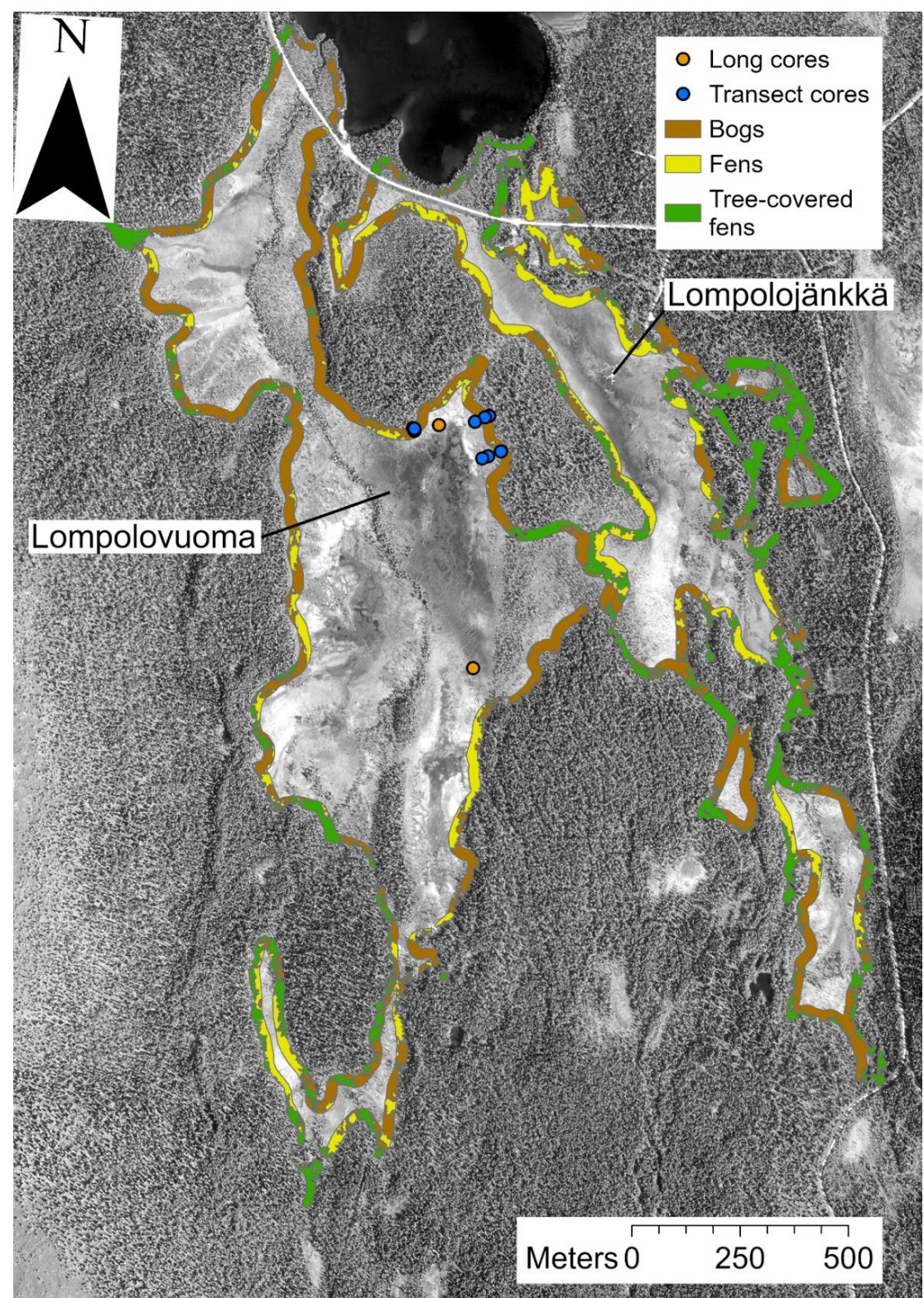

380

**Figure 6. Peatland margin vegetation communities. The area 25 meters from the peatland margin is divided into bog-type (brown), fen-type (yellow), and tree-covered fen type (green) in Lompolovuoma study basin and adjacent Lompolojänkkä basin. In addition, location of the study transect peat cores (blue circles) and long cores (orange circles) are shown. This figure contains data from National Land Survey of Finland NLS Aerial photographs database.**

**Table 2. The vegetation class coverage and peatland area. Table shows the total area of the Lompolovuoma and Lompolojänkkä peatland basins, and proportion of 3 vegetation community classes in the peatland basins: Bog-type, fen-type, and tree-covered fen type. In addition, the total area of the 25-meter margin, and proportions of the vegetation community classes is shown. In the final 2 columns, the proportion of the vegetation community classes is shown individually for Lompolovuoma and Lompolojänkkä basins.**

| Vegetation class | Peatland | | Peatland margins | | | |
| --- | --- | --- | --- | --- | --- | --- |
| | Total area, ha | Total area, % | Margin area, ha | Margin area, % | Lompolo-vuoma | Lompolo-jänkkä |
| Bog-type | 47.4 | 34 % | 19.2 | 44 % | 54 % | 35 % |
| Fen-type | 64.8 | 46 % | 10.3 | 23 % | 20 % | 26 % |
| Tree-covered fen | 29.0 | 21 % | 14.4 | 33 % | 26 % | 39 % |
| Total area | 141.2 | 100 % | 43.9 | 100 % | 100 % | 100 % |

**3.6 Hydrological analyses**

The simulated groundwater – surface water (GW-SW) exchange patterns for the current climatic and groundwater table (GWT) elevation change are shown in Fig. 7a and 7b, respectively. The calculated areas by contour group and peatland vegetation group are presented in Table 3 for the GW-SW exchange fluxes and in Table 4 for the changes in terms of GWT elevation.

In terms of exchange flux, the areas classified as fens indicate the dominance of the GW exfiltration over infiltration processes in the simulations. In contrast, the bog areas indicate more balance between infiltration and exfiltration processes with a slight prevalence of the infiltration area. The areas classified as treed fens show the dominance of infiltration. However, ~30 % of the total treed fen area is in the vicinity of the ditch network (the rightmost part of the peatland system), which impacted the peatland vegetation as indicated by aerial photos (National Land Survey of Finland, 2023). After excluding the drained areas from treed fens, the GW-SW exfiltration pattern is more balanced with a slight prevalence of exfiltration.

In terms of groundwater table elevation changes, the simulated drier climatic conditions have a mild impact on the areas classified as fens, with 59 % of the water table decreasing by less than 1cm and 89 % by less than 5 cm. In contrast, the areas classified as bog are more susceptible to GWT changes. They are characterised by significantly less extent of the areas with mild (less than 1 cm and 5 cm) table decrease (only 28 % and 70 % respectively) and a significant portion (30 %) with a substantial decline (more than 5 cm). The treed fen areas, excluding ditches, suggest that the water table decrease would be variable, with more GWT reduction than in the case of open fens but lower than in the case of bogs.

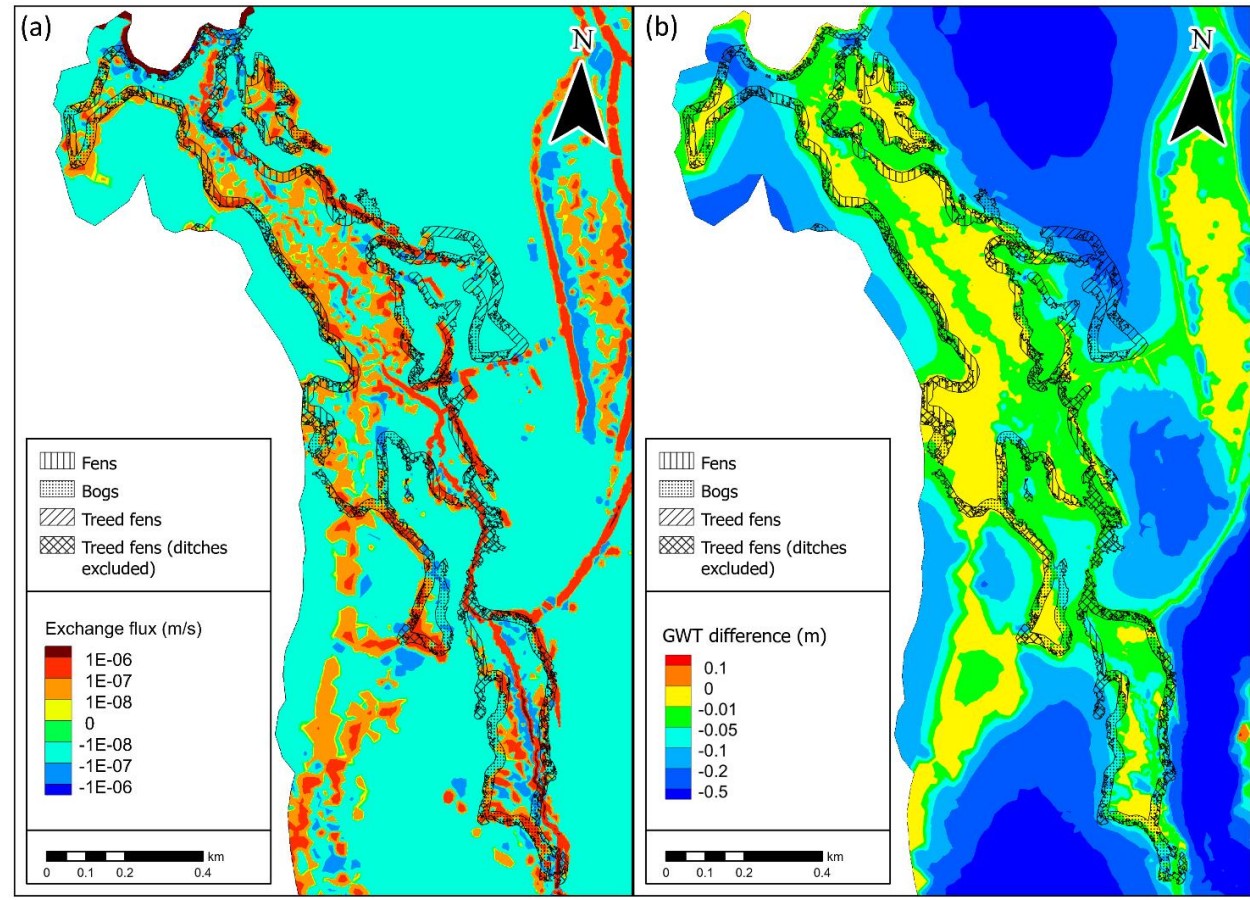


**Figure 7a. The GW-SW exchange flux patterns from Lompolojänkkä sub-basin averaged for summer 2017 representing**
**the current climate conditions. Positive flux values indicate the locations of groundwater exfiltration and infiltration**
**towards groundwater. 7b. The groundwater table elevation changes result from the drier climate conditions. Negative**
**values indicate the groundwater level decrease and positive increase.**

**Table 3. Exchange flux contour areas by vegetation type. Treed fens (ditches excluded) exclude the open drainage areas.**

| Exchange flux (m/s) | | | Area (m$^2$/ %) | | |
|---|---|---|---|---|---|
| Min value | Max value | Fen | Bog | Treed fens | Treed fen (ditches excluded) |
| | <-1E-6 | 0 / 0 % | 18 / 0 % | 0 / 0 % | 0 / 0 % |
| -1E-06 | -1E-07 | 1471 / 2 % | 2818 / 4 % | 3130 / 4 % | 2360 / 4 % |
| -1E-07 | -1E-08 | 18822 / 30 % | 34813 / 47 % | 46693 / 55 % | 26342 / 45 % |
| -1E-08 | 0.00 | 2936 / 5 % | 3662 / 5 % | 2549 / 3 % | 2256 / 4 % |
| 0.00 | 1E-08 | 3067 / 5 % | 3603 / 5 % | 2661 / 3 % | 2388 / 4 % |
| 1E-08 | 1E-07 | 23583 / 38 % | 19196 / 26 % | 12452 / 15 % | 10989 / 19 % |
| 1E-07 | 1E-06 | 12529 / 20 % | 9859 / 13 % | 16696 / 20 % | 14106 / 24 % |
| 1E-6> | | 83 / 0 % | 85 / 0 % | 555 / 1 % | 552 / 1 % |

| | | | | |
|---|---|---|---|---|
| **Total negative flux** | 23229 / 37 % | 41311 / 56 % | 52373 / 62 % | 30957 / 52 % |
| **Total positive flux** | 39262 / 63 % | 32743 / 44 % | 32363 / 38 % | 28035 / 48 % |
| **Total flux** | 62491 | 74053 | 84736 | 58993 |


**Table 4. Water table difference between baseline and drier climates by contour areas and vegetation type. Treed (ditches excluded) exclude the open drainage areas.**

| Water table decrease from the baseline model (m) | | Area (m$^2$/ %) | | | |
|---|---|---|---|---|---|
| **Min value** | **Max value** | **Fen** | **Bog** | **Treed fens** | **Treed fen (ditches excluded)** |
| **-0.2** | -0.5 | 15 / 0 % | 642 / 1 % | 8084 / 10 % | 921 / 2 % |
| **-0.1** | -0.2 | 1496 / 2 % | 7855 / 11 % | 11539 / 14 % | 4043 / 7 % |
| **-0.05** | -0.1 | 5531 / 9 % | 13195 / 18 % | 11111 / 13 % | 6479 / 11 % |
| **-0.01** | -0.05 | 18608 / 30 % | 31348 / 42 % | 35656 / 42 % | 31314 / 53 % |
| **0** | -0.01 | 36846 / 59 % | 21040 / 28 % | 18428 / 22 % | 16317 / 28 % |
| **Total** | | 62495 | 74080 | 84818 | 59074 |

**4. Discussion**

We studied lateral expansion and vegetation succession of peatland margins in a subarctic Lompolovuoma fen in Finnish Lapland. Our results show that the studied margins in Lompolovuoma started to develop ca. 2000 years ago and the youngest basal age of 390 cal BP was located still a few meters from the current forest edge. Peat initiation in the margins occurred in several independent loci that only later coalesced into a continuous peatland. The initial wet *Cyperaceae – Ericaceoae* marginal communities inhabited the fen margins over the time periods reaching from a few centuries to a millennium and the following establishment of *Sphagnum* moss dominated communities was asynchronous. Starting from the end of the 1800[th] century, these margins shifted to a climax bog plant community characterized by hummock sphagna and ericaceous vegetation. This change occurred mostly over a remarkably short time, in a few decades. However, our data also showed that forest vegetation had on several occasions intruded to the already established peatland, suggesting an on-going "power-struggle" between peatland and forest ecosystems. The marginal "bog-type" vegetation currently covers roughly 50 % of the margins in the Lompolovuoma sub-basin, while in the adjacent sub-basin of Lompolojänkkä, only 35 % have reached the ombrotrophic stage. Our hydrological GW-SW model indicates that in the "fen-type" margins high water-tables are sustained even during dry climatic conditions, showing a resistance-potential to fen-to-bog transition.

**4.1 Non-linear development of peatland margins in Lompolovuoma fen**

The formation of Lompolovuoma peat margins investigated here began ca. 2200 years ago. This differs from the central part of the mire, where limnotelmatic Equisetum peat found at the bottom the long peat profiles suggest that peat formation initially occurred over water body. Similar to the results by Juselius-Rajamäki et al. (2023), these data

contradict the traditional perception that peatland expansion has ceased or markedly slowed down during the last 2000
years in Fennoscandia (Ruuhijärvi, 1983; Sjörs, 1983). Rather, the current findings suggest this presumption is due to
under-representation of studies and sample collection from the mire marginal areas rather than an actual ceasing of
lateral expansion (Kuhry & Turunen, 2006; Ruppel et al., 2013). In transects 2 and 3, the expansion of new peat
surfaces occurred from individual miniature loci evidenced by the oldest basal ages found from the middle of the
profiles, while in transect 1 the oldest basal age was acquired for the profile closest to the main mire (Fig. 2a-c).
However, the basal age and the basal elevation of the T1Sp matches closely to the age and elevation of the oldest
bottom age of transect 2, suggesting a relatively simultaneous initiation process.

The basal ages from the studied transects show that after the initial peat formation, the individually formed peat patches
spread both downhill towards the main mire area, and uphill towards the adjacent forest. Only later, separate peat
patches were connected to main mire basin. Such convergence of the multiple smaller loci to a single peatland mass
has been reported both during the early Holocene (Almquist-Jacobson & Foster, 1995; Korhola, 1992, 1994;
Mathijssen et al., 2014, 2017) and for more recently developed mire margins (Juselius-Rajamäki et al., 2023).
However, the mechanisms behind the development of individual peat patches and the later convergence have received
only little attention and remain unresolved (Noble et al., 1984).

In Lompolovuoma, peat initiation occurred in steep slopes on average exceeding 0.5°, a threshold known to restrict
peat formation in more continental regions where availability of water is not excessive (Almquist-Jacobson & Foster,
1995; Loisel et al., 2013; Zhao et al., 2014). Thus, in the past, suitable conditions promoting the initiation of individual
peat patches must have existed. The peat patches may have started to form in small topographical depressions that,
although initially well-drained, may become impervious due to deposition of organic or fine inorganic matter,
formation of hardpans in the Spodosol layer, or deposition of ash due to forest fires, creating favourable conditions
for peat formation (Klinger, 1996; Le Stum-Boivin et al., 2019; Mallik et al., 1984; Noble et al., 1984; Rydin &
Jeglum, 2013). No full-scale subsoil topography measurements were conducted, but field survey data did not reveal
any clear depressions underlying any of the oldest peat profiles. Another scenario is, that under sufficiently humid
conditions the peat formation began directly on the steep slopes, as suggested for southern Finland peatlands (Korhola,
1996). Climate reconstructions suggest wet climate phase prevailed in Lapland between 2500 and 2000 BP (Eronen
et al., 1999; Luoto & Nevalainen, 2015), which may have promoted peat formation even in a relative steep slope, such
as presented here.

The vertical growth of peat as a driving mechanism for lateral expansion has been traditionally linked to raised mires
(Foster & Wright, 1990). However, although the shape of the Lompolovuoma surface has remained concave, the low
hydraulic conductivity of saturated peat (Ingram, 1978; Rydin & Jeglum, 2013) combined with the large amounts of
waters flowing from surrounding uphill areas, especially during the snow-melt period (Autio et al., 2023) could
nevertheless cause flooding in suitable locations even if these locations were separated from the main mire body.
Similarly, previous studies have shown that although no elevated mire centre exists, significant lateral expansion of
peatland has occurred (Almquist-Jacobson & Foster, 1995; Korhola, 1994, 1996; Korhola et al., 2010; Mathijssen et
al., 2017), suggesting that even on flat or concave shaped peatland basins peat accumulation can lead to redistribution
of waters towards mire margins. Low-severity fires in adjacent forests are also known to promote peatland lateral
expansion, as the reduced tree-cover decreases evapotranspiration and promotes colonization of *Sphagnum* due to
increased light availability (Le Stum-Boivin et al., 2019; Novenko et al., 2021). However, in our basal layers only a
single charred wood piece used for dating was found, while microscopic analysis of the basal layers did not reveal any
charcoal (Fig. A2, A3, A4). Thus, forest fires did not likely play an important role in the peat initiation in question.

**4.2 Autogenic and allogenic drivers behind the plant community succession**

The initial *Cyperaceae-Ericaceae*-dominated stage found in our study site is commonly present in the basal layers of
the peatland margins in Finland (Juselius-Rajamäki et al., 2023; Mathijssen et al., 2017). On the other hand, many
studies have shown that sphagna is frequently found in the first stages of the paludification process (Le Stum-Boivin
et al., 2019; Noble et al., 1984; Rydin & Jeglum, 2013). This variation can also be seen in our comparison profiles, as
the margin of Syysjärvi site shows similar development as in Lompolovuoma, while in the more southern Salamajärvi
and Patvinsuo *Sphagnum* mosses were already present during the initial paludification (Fig. A1). The lack of
*Sphagnum* mosses in Lompolovuoma margin during the peatland initiation is likely explained by the hydrological
conditions. At the onset of the peatland expansion, the water table was likely fluctuating, as shown by the presence of
both forest mosses and mycorrhizal fungi *Cenococcum geophilium* (van Geel, 1978) linked to dry conditions, and
discovered cyperaceous vegetation, for example *Carex limosa* (Fig A3, A4) usually referring to relatively wet
hydrological regime (Visser et al., 2000). *Sphagnum* mosses require constantly humid conditions for colonization
(Fenton et al., 2007; Sundberg & Rydin, 2002), and even though they can tolerate limited periods of desiccation (Hájek
& Vicherová, 2014), the prolonged fluctuating water sources in margins likely prevented early colonization by
*Sphagnum* mosses. Only after the gradual development of mire conditions proper in the margins, was the spread of
the peat mosses possible.
After the initial C-E stage, colonization of sphagna occurred asynchronously between 670 and 1770 CE. This gradual
transition towards mixed *Cyperaceae-Ericaceae-Sphagnum* vegetation was likely moulded by autogenic development
as changes driven by allogenic forcing would occur over large areas within a relatively short time span rather than
over a millennium, as discussed in Väliranta et al. (2017). This conclusion is supported by the fact that no evidence
of forest fires was found in the peat profiles. Similarly, no such contemporary climate event has been detected which
could promote large scale changes in vegetation and simultaneous spatial colonization of sphagna (Hanhijärvi et al.,
2013; Linderholm et al., 2018; Luoto & Nevalainen, 2015). The comparison profiles from Patvinsuo and Salamajärvi
also show gradual increase in the *Sphagnum* mosses, albeit at much shorter time scale than witnessed in
Lompolovuoma, while in Syysjärvi the shift to *Sphagnum* moss dominance was extremely rapid (Fig. A1).
Although the decomposition of the bottom-most layers of peat prevented complete species-level identification of
cyperaceous vegetation, increasing number of *Eriophorum vaginatum* remains were found in layers preceding the
*Sphagnum* colonization (Fig. A2, A3, A4). Like *Sphagnum* mosses, tussock-forming cyperaceous vegetation may act
as 'ecosystem engineers' (Palozzi & Lindo, 2017; Väliranta et al., 2017) and the importance of *Eriophorum vaginatum*
facilitating the fen-to-bog transition has been recognized in various studies (Hughes, 2000; Hughes & Dumayne-Peaty,
2002; Väliranta et al., 2017). These species can alter local conditions, such as hydrology and acidity (Hughes, 2000;
Hughes & Dumayne-Peaty, 2002) and produce litter highly resistant to decay, thus promoting peat accumulation
(Wein, 1973). This accumulation process can be further amplified by presence of ericaceous vegetation (Hughes,
2000). Although in the studied margins the accumulation of the peat during C-E stage was modest, elevated surface
combined with increased acidity seems to have been sufficient to create conditions suitable for establishment of
*Sphagnum* species found in the studied margins, likely protecting them from alkaline waters and complete inundation
known to impede colonization sphagna (Granath et al., 2010; Ruuhijärvi, 1983; Sallantaus, 2006).

After the colonization, *Sphagnum* mosses accelerate the change in local conditions (Rydin & Jeglum, 2013), increasing
their competitiveness against other mire vegetation and leading to ombrotrophication. In some cases, this change can
occur rapidly (Tahvanainen, 2011) and synchronously over wide area (Loisel & Bunsen, 2020), while also more
gradual changes have been observed (Väliranta et al., 2017). In Lompolovuoma, initially, the abundance of sphagna
remained low after the first establishment, but a more dramatic change occurred towards the end of the 19th century,
when *Sphagnum* mosses started dominating the margin plant communities, and most of the cyperaceous vegetation
disappeared, leading to the current S-E vegetation stage. This change coincided with the end of the 'Little Ice Age'
(LIA), when humid and cool climate conditions were followed by increasingly warm temperatures (Hanhijärvi et al.,
2013). Similar post-LIA fen to bog shifts have been reported in previous studies where data is captured from central
parts of the peatland (Granlund et al., 2022; Kolari et al., 2022; Loisel & Yu, 2013; Magnan et al., 2018; Piilo et al.,
2019; Primeau & Garneau, 2021; Robitaille et al., 2021), while our results show similar recent changes occurring in
the margins. Current results are supported by a study from adjacent Lompolojänkkä basin showing a similar kind of
recent vegetation shift in the margins (Kuuri-Riutta et al., 2024) and those of our comparison profile from Syysjärvi
(Fig. A1). Thus, although aapa mires are generally described as having wet central parts and dryer margins, our results
show that dryer margins supporting sphagna may have formed rather recently.

Although these recent fen-to-bog transitions have occurred during dry climatic conditions post-LIA, similar shift has
also occurred during wet climate phases (Väliranta et al., 2017), as the only requirement for the process is the
separation of the peat surface from the groundwater supply (Hughes, 2000; Hughes & Barber, 2003). During wet
climatic conditions, the accumulation of peat is promoted, rather high water-table levels are maintained, and the fen-
to-bog transition leads to a bog pool and lawn communities (Hughes & Barber, 2003). On the other hand, dry climate
conditions decrease the water-table, that enables the species with tolerance towards drought or fluctuating water-tables
to out-compete other species (Hughes & Barber, 2004). In Lompolovuoma margins, hummock-forming *Sphagnum*
species, especially *Sphagnum fuscum*, increased markedly during the ultimate shift to ombrotrophic bog conditions.
The final fen-to-bog transition in the studied mire margins appears to be caused by the drier and warmer climate, as
only sporadic presence of non-hummock *Spaghna* was detected in the peat profiles (Fig. A2, A3, A4). Moreover, the
most marginal peat profiles in transect 1 and transect 2, as well as in comparison profile from Patvinsuo, show that
the peatland vegetation has been replaced completely by forest vegetation on several occasions. This suggests that
peatland expansion may be reversed at least temporary.

However, based on the remote sensing data, similar ombrotrophication has not occurred across all margins in
Lompolovuoma and adjacent Lompolojänkkä basins. The ombrotrophic S-E stage can currently be found roughly in
50 % of the margins of the Lompolovuoma basin while this stage has been reached only in ca. 35 % in adjacent
Lompolojänkkä. Similarly, the central part of adjacent Lompolojänkkä basin has shown no evidence of fen-to-bog
transition (Mathijssen et al., 2014), but transition is ongoing in the margins (Kuuri-Riutta et al., 2024). Thus, it appears
that for the transition from fen to bog to occur, certain prerequisites and conditions must be met. Our hydrological
model, based on the Lompolojänkkä basin, showed that while marginal fens were generally ground-water recipients,
the bog-type vegetation acted preferentially as surface water infiltration areas. By decreasing the effective precipitation
in the hydrological model to mimic dryer conditions, the highest levels of water table drawdown were found in the
current "bog-type" margins, marking these locations more likely to suffer drying conditions. Although both the
analysis of vegetation cover (Räsänen et al., 2021) and hydrological model (Autio et al., 2023) contain some degree
of uncertainty, the application of the hydrological model over the marginal peatland types supports our hypothesis of
drop in ground-water levels as a likely cause for the final shift towards ombrotrophic climax stage.

**4.3 Implication for carbon balance and future trajectories of vegetation succession in aapa mire margins**
It has been shown that in fen conditions climate forcing from peatland complex can remain positive (e.g., climate
warming effect) for most of the development history due to high methane emissions and only after continuous carbon
uptake and expansion of bog vegetation the climate forcing turns negative (Korhola et al, 1996; Mathijssen et al.,
2017, 2022). For example, in adjacent Lompolojänkkä basin the modelled climate warming effect persisted up to 2000
years (Mathijssen et al., 2014). Thus, it is likely that during the initial minerotrophic C-E stage lasting between 150
and 1250 years, the mire margin had a climate warming effect. Afterwards, a shift to decay-resistant *Sphagnum*
vegetation, lower water table leading to reduced methane emissions and continuous carbon uptake would likely have
the same effect. Decrease of the cyperaceous vegetation especially during the last ca. 100 years would have reduced
the methane emissions even further (Bubier et al., 1993; Ward et al., 2013). Although our study did not include carbon
balance calculations, the shift towards bog community on the studied margins suggest that under current conditions,
the margins would likely proceed to have a climate cooling effect. However, drying trend detected in the European
peatlands (Swindles et al., 2019) could also turn these locations to carbon sources, if sufficient moisture conditions
are not retained (Zhang et al., 2020).

As this study and studies by Juselius-Rajamäki et al. (2023) and Kuuri-Riutta et al. (2024) show, new peatland areas
are currently widely being formed in the mire margins all over subarctic and boreal zone under natural conditions.
However, in many places this development has been blocked by the ditching of mire margins (Sallinen et al., 2019),
while the widespread drying of peatland surfaces during the last ca. 300 years may suggest that detrimental climatic
conditions for lateral expansion are forming (Swindles et al., 2019). In addition, as revealed by this study, the
succession of mire margins even in the same peatland can differ, with some margins retaining their initial wet
minerotrophic characteristics, while others develop to ombrotrophic bogs. Due to the opposite climate forcing, the
effect of this recent mire expansion on the climate depends on the scope of different peatland types across new mire
margins and their later development. The knowledge on the developing peatland margins and their plant community
succession still remains scarce. As the lateral expansion of peatlands has had a significant effect on atmospheric
greenhouse gas concentrations in the past (Korhola et al., 2010; Peng et al., 2024), we suggest that more studies across
the northern peatland margins are needed to reveal the wider effect of this recent lateral peatland expansion on the
global carbon budgets.

## 5 Conclusions

Our research shows that the studied mire margin in Lompolovuoma basin has continued to increase in area since ca.
2000 cal BP, but this development has not progressed linearly. Rather, the current mire margin has formed from
several individual loci and via patches that have merged as the local hydrology has transformed suitable for peat
formation. After the initial wet "fen-type" conditions, that persisted for markedly long period, colonization by
*Sphagnum* mosses, the change to current "bog-type" conditions represents a remarkable swift shift. This change was
driven by dryer climatic conditions following the LIA as shown by our hydrological model. However, not all margins
in Lompolonvuoma and Lompolojänkkä basins have shifted to "bog-type" communities suggesting that wetter "fen-
types" are at least partially resistant to hydrologically driven regime shifts. This study shows that even on the basin-
scale, peatland margins are highly heterogeneous systems, and this should be taken into account when assessing the
effects of past and future lateral expansion trend on the peatland area and peatland carbon dynamics.

**6 Appendices**

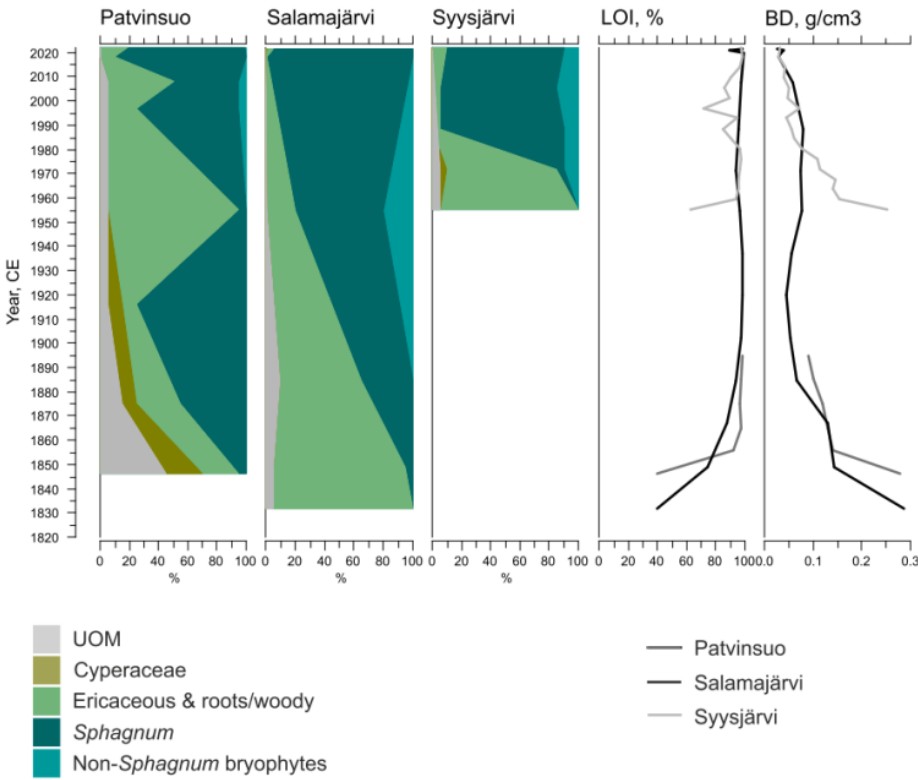


**Figure A1. Fossil plant records (left) including undetected organic material (UOM), and loss on ignition (LOI) and bulk density (BD) for supplementary profiles. Proportion of vegetation type and LOI in percentages (%), unit for bulk density is g/cm³**

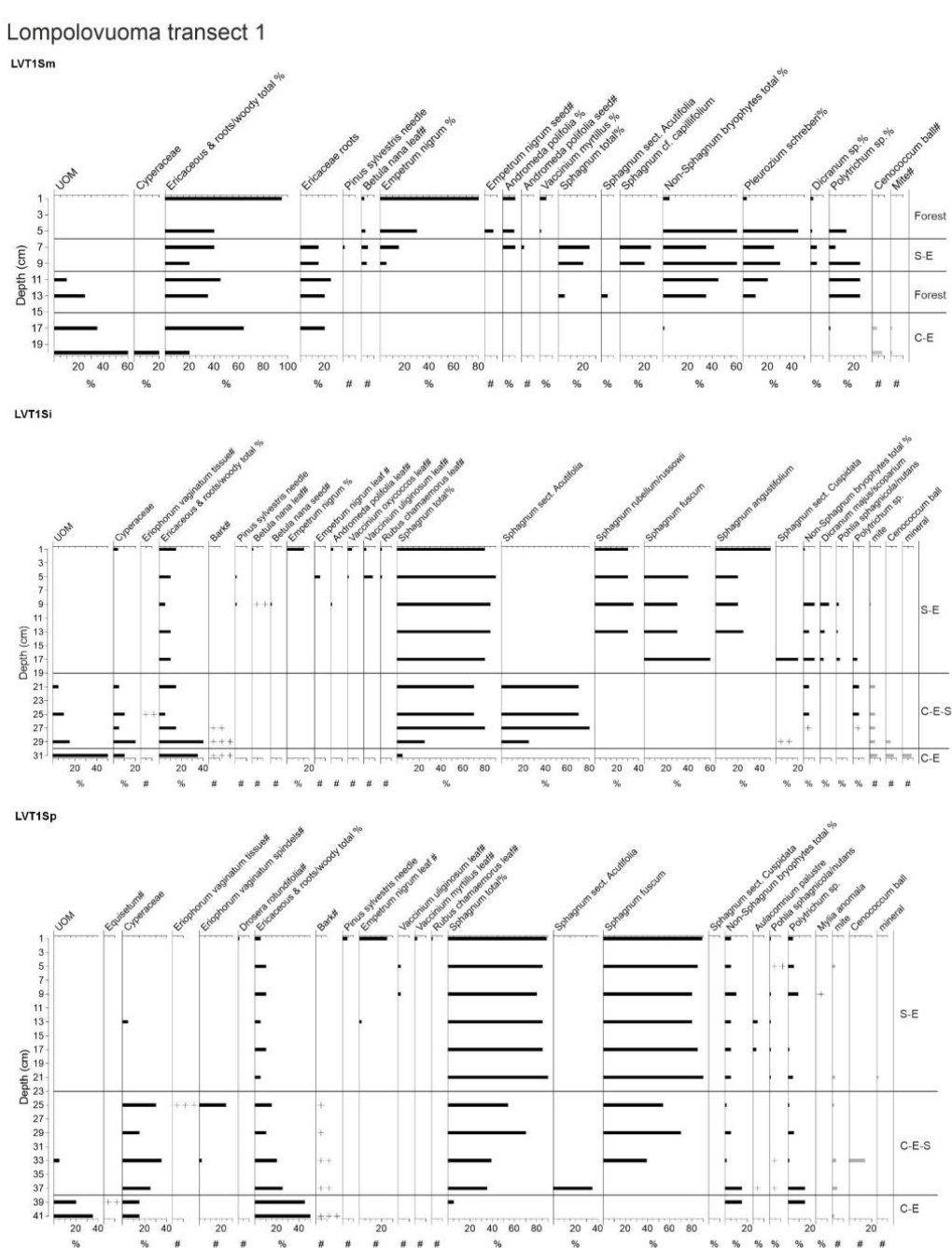


**Figure A2. Macrofossil data for peat profiles in transect 1. Unrecognized organic matter (UOM), and plant species or species group are presented as well as remains of mites, *Cenococcum*, and mineral content.**

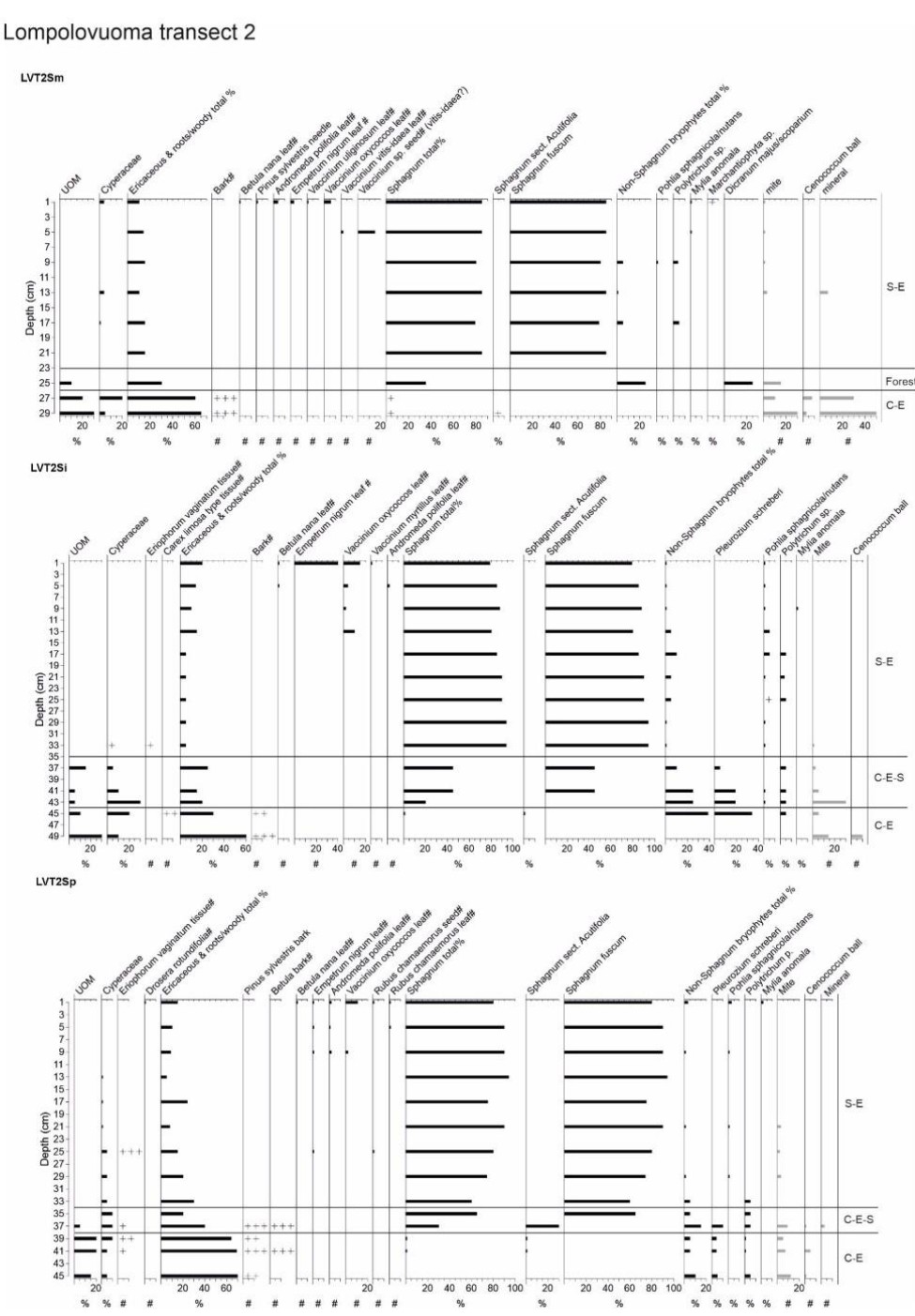

Figure A3. Macrofossil data for peat profiles in transect 2. Unrecognized organic matter (UOM), and plant species or species
group are presented as well as remains of mites, *Cenococcum*, and mineral content.

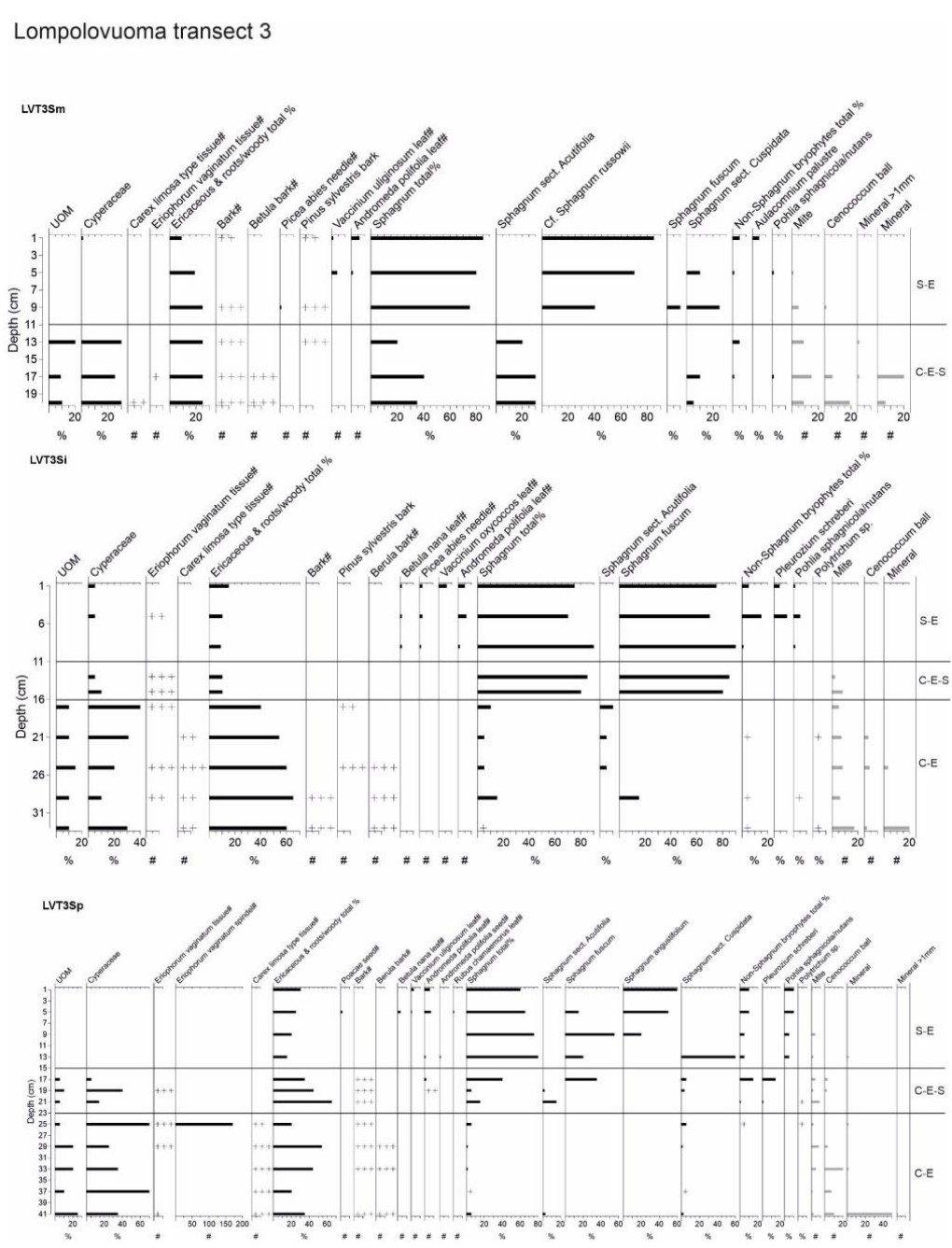


**Figure A4. Macrofossil data for peat profiles in transect 3. Unrecognized organic matter (UOM), and plant species or species**
**group are presented as well as remains of mites, *Cenococcum*, and mineral content.**


## 7 Data availability statement

The data for peat properties, peat core locations and age-depth models are available on the figshare: https://doi.org/10.6084/m9.figshare.25941493.v1

## 8 Author contributions

T.-J-R and M.V. conceived the idea for the article. T.J-R, E.T., M.V., A.A., H.M., and P.A-A. collected the field data. T-J.R. and S.P. performed the macrofossil analysis. T.J-R. and S.S-P. conducted the $^{210}$Pb-analysis. T.J-R. conducted the spatial analysis. A.A., H.M., and P.A-A. conducted hydrological modelling. T-J.R. created the initial draft for the manuscript. All authors contributed to the drafts and gave the final approval for publication.

## 9 Conflict of interest statement

The authors declare that they have no conflict of interest.

## 10 Acknowledgements and financial support

T.J-R. was funded by Tellervo ja Juuso Walden foundation, M.V., S.P. and T.V. received funding from - Research council of Finland project 338631 and 349193. Groundwater modelling and GPR dataset as part of University of Oulu activities were supported by Research Council of Finland ACWI project (project nro 316349), Freshwater Competence Centre (FWCC) and DIWA-flagship. We acknowledge the support from the Ministry of Transport and Communication through ICOS Finland and from the WetHorizons project (Horizon Europe GAP-101056848).

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
