# Peer review of "External and internal drivers behind the formation, vegetation"

_EGUsphere, 2024_

## Author Response (AR1)

**REFEREE 1**

**The reviewed manuscript is well-written and explores the interesting topic of lateral expansion of subarctic mires. The interpretations are based on 3 transects, each with 3 short peat profiles. Moreover, the representation of three main vegetation types in the mire margins was explored by remote sensing method and the hydrological processes were studied by hydrological modelling. Generally, I like the study and its results, but I have some comments and questions, which could point to some problematic or unclear parts of the research and could be discussed in more detail. I suggest publishing this manuscript after Minor revision.**

We thank the referee for the acute suggestions and comments on our article. These comments will improve the article and make the contents more easy and accurate for the readers. We also thank for the favourable view on the article on general. In this document, we have addressed all the topics raised by the referee.

Below, we answer point-by-point to the comments and questions. The comments by the referee have been bolded and colored violet, while our answers have been colored green. The journal instructions stated that at this point no changes to the actual manuscript should be made. We have added the suggestions for the improved text after each answer (underlined) whenever necessary and if no further improvements are needed, these will be added to the manuscript. Similarly, the minor and technical corrections will be addressed in the new manuscript version.

**General comments and questions**:

**Line 96: I would like to ask how the places for transects were selected and especially how the distances between particular profiles within the particular transects were assessed. I noticed in figures 2a-c, that the length of transects differed and distances among particular profiles as well. This information could be added.**

The places of the transects themselves were selected to represent the "bog-type" mire margin area that was accessible at the sampling time. Several adjacent transects were collected as prior studies (see for example Juselius-Rajamäki et al. 2023) have shown that single transects are not able to capture the variability in the mire margin development even within a small area.

The variability of the length of the "bog-type" margin in studied margin largely controlled both the length of the transects and the sampling locations. All profiles were collected from lawn-microform with similar types of vegetation. The peat profile collected closest to the mire center for each transect was collected at the end of the transect just inside the "bog-type" margin. The other samples were collected along the transect following the above-mentioned criteria, but no set distances were used between the samples due to variable lengths of the transects and variability of the landscape.

For better comparison value, in future we aim to use equidistant approach for sampling procedure whenever possible. However, generally in the field conditions don't allow exact placement of the samples and similar more or less "flexible" placement is conducted instead.

To clarify the placement of the transects and the sampling locations, we will add following information to line 151 in methods section:

The transects were placed on the "bog-type" margin with variable width, which led variable lengths in our transects. All profiles were collected from lawn-microform with similar types of vegetation. The first sampling location for each transect was at the extreme end of the "bog-type" margin next to the central

fen area. The other samples were collected along the transect from locations fulfilling the above criteria, and in such manner that entire length of "bog-type" -margin at the transect location was covered.

**Line 158: It should be added a reference to the LOI method.**

We will add the following reference to a LOI method: Heiri et al. (2001). Heiri, O., Lotter, A. F. & Lemcke, G. (2001). Loss on ignition as a method for estimating organic and carbonate content in sediments: Reproducibility and comparability of the results. Journal of Paleolimnology, 25(1), 101-110. https://doi.org.10.1023/A:1008119611481

**Line 162: "limnotelmatic *Equisetum* peat" – does it mean that there are shallow lakes or aapa flarks at the bottom of the mire? Maybe this could be more explained in the manuscript.**

The central part of the current fen has most likely been a shallow lake, or slow-moving river at some point of time, while the current fen center consists of string and flarks. Although not strictly within the scope of the study we understand that this background information could be useful to understand the history of the mire basin. We will include following text starting from line 444:

The formation of Lompolovuoma peat margins investigated here began ca. 2200 years ago over mineral soils. This differs from the central part of the mire, where limnotelmatic Equisetum peat found at the bottom the long peat profiles suggest that peat formation initially occurred over water body.

**Line 203-204: I don't fully understand what is meant by "prior values" and how they were modified. Maybe it is a matter of terminology. Could it be better explained?**

Prior value(s) are used in models using Bayesian statistics such as Age-depth model developed by Blaauw and Christen (2011). In Bayesian statistics prior is used to represent the initial assumption of the data, generally based on expert opinion or previous data collected, in this case peat accumulation rates from similar vegetation composition from similar sites. These studies are cited in line 202-203. However, as the vegetation composition analysis does not give exact correct data but is based on the analyzation by individual the prior values need to be modified to ensure the model fit (e.g., increase or decrease the used prior accumulation values). It should be noted that the prior values were altered if the model fit (the 'fit' or 'energy' of all MCMC (Markov Chain Monte Carlo) iterations of the run) was not satisfactory. If the model fit was OK (as described in: https://cran.r-project.org/web/packages/rbacon/vignettes/rbacon.html#running-rbacon) then no changes were made. In plain English, the prior data was not changed to "suit our needs".

We will add and alter the method description in line 203 as follows:

We used these accumulation rates as a prior value for the age-depth models for corresponding vegetation community stages. After initial model run, if the model fit was not satisfied (Blaauw & Christen 2011), the prior values were altered to ensure the model fit.

**Table 1: Looking at Table 1 with all radiocarbon dates, one question came to my mind. The material used for dating in the basal sample was, in many cases, wood. The difference between the age of the basal (wood) sample and the next sample is mostly high (about 1000 years), even if the difference between these two samples is sometimes only a few centimetres e.g. T1Si 30-31 cm is dated 1250 BP and T1Si 27-28 cm 315 BP (935 years in 2-3 cm). Didn´t you consider the presence of a hiatus? Or cannot be another interpretation that the age of basal wood is the age of woodland before lateral spread started? Did you identify the wood? Was it tree or dwarf shrub wood? This is unclear to me. Moreover, you interpret the initial basal stage of mire as a wet stage. Why was the peat accumulation so low in the wet stage? I would expect the opposite, quick**

**accumulation of peat. As you write on lines 294-295, the peat in this stage was strongly decomposed; it would also indicate a rather dry stage, not wet.**

We used small pieces of woody material for dating, most of which were remains of dwarf shrubs, but in some cases also pieces of birch. However, no further species identification was conducted. We don't consider the possibility for these pieces to represent the woodland likely, as aerobic decay would decompose such pieces rather quickly.

As you suggest, the hiatus could have occurred between the initiation and the upper limit of the C-E layer, but we have no means of discovering whether this actually occurred.

It should also be noted that although the difference in cm is currently only a few centimetres, there has been hundreds of years of decomposition affecting the peat layers. What this means is that the actual amount of peat accumulated is not, in fact, 2-3 cm, but 2-3 cm plus all the material that has accumulated and later decomposed. We recognize that the effect of decomposition of the older peat layers has not been properly stated in the manuscript and will clarify this accordingly:

Lines 212, 215-217:
Text on vertical peat increment is removed

Lines 313-315:
At the onset of peat development in the mire margins, the C-E vegetation community dominated (Fig. 2a-c, Fig. 3-5). In transects 1 and 2 this layer is currently thin, only up to 4 cm in transect 1 and from 5 cm to 9 cm in transect 2. In the transect 3 the C-E layer is thicker, 16 cm in T3Si and 18 cm in T3Sp.

Lines 368-371 & Figure 6:
We will remove this paragraph. As notified by other referee, comparison between peat accumulation should incorporate a decay model, which we currently are lacking. Although this would likely reveal some interesting results about the rate of vertical peat accumulation, this is not the main focus of the manuscript. Manuscript is also already quite long.

Lines 574-575: We will also remove the mention of peat accumulation rates from this study from lines 574-575, and justify the discussion by existing results from other studies by altering the paragraph as follows:

It has been shown that in fen conditions climate forcing from peatland complex can remain positive (e.g., climate warming effect) for most of the development history due to high methane emissions and only after continuous carbon uptake and expansion of bog vegetation the climate forcing turns negative (Korhola et al., 1996; Mathijssen et al., 2017, 2022). For example, in adjacent Lompolojänkkä basin the modelled climate warming effect persisted up to 2000 years (Mathijssen et al. 2014). Thus, it is likely that during the initial minerotrophic C-E stage lasting between 150 and 1250 years, the mire margin had a climate warming effect. Afterwards, a shift to decay-resistant Sphagnum vegetation and lower water table leading to reduced methane emissions would likely have the same effect. Decrease of the Cyperaceous vegetation especially during the last ca. 100 years would have reduced the methane emissions even further (Bubier et al., 1993; Ward et al., 2013). Although our study did not include carbon balance calculations, the shift towards bog community on the studied margins suggest that under current conditions, the margins would likely proceed to have a climate cooling effect.

We further discuss the C-E stage wetness contradictory in your comment for lines 497-498 (524-525).

Line 302: What do you mean by "varying degree of *Ericaceous* vegetation"? Percentages or changing species composition?

We mean percentage (%). The text will be modified as follows to clarify it: A varying amount (%) of Ericaceous vegetation is usually mixed with the Sphagna.

**Line 365-366: Do you think the lateral spread was regular? As you discussed later, the separate small peat bodies were later connected with the main mire body. Thus, the question arises, does calculating the lateral expansion rate make sense?**

These rates represent the average lateral expansion rates between two points calculated from the basal ages, but no assumption whether this is constant or not is included. Indeed, we think it highly improbable that the rate was constant. The idea for presenting these rates is to give comparable data to similar studies that collect transect data measuring the peatland expansion. In this case, the lateral expansion occurred from the peat patches towards the main mire and the mire proper, which can be considered as lateral expansion of the peatland area although not directed entirely outwards. To make this more clear, we will add word average to the lines 365 and 366.

**Line 409: I would like to ask if the modelling of groundwater table elevation changes is based on the recently measured water tables. It is not clear to me.**

Pallaslompolo catchment has several on-going continuous water table measurement points in peatlands and mineral soil groundwater. Hydrogeosphere (HGS) model was calibrated and validated against these measurements. More details from model calibration can be found from: Autio et al. 2023. Journal of Hydrology. https://www.sciencedirect.com/science/article/pii/S0022169423012842

**Line 440-441: What could be the reason that some mire margins stayed wet? Couldn´t be the stronger discharge of groundwater the reason? How much the mire-margins are influenced by groundwater? Did you measure the pH and conductivity of groundwater to assess the mineral richness of the groundwater? It could be important for *Sphagnum*'s establishment.**

The sentence in lines 440-441 at the beginning of the discussion does not go fully to detail the causes why some margins were resistant to change. This is discussed in lines 561-569, which states that this effect is indeed caused by the fact that these wet margins were generally ground-water recipients. Based on the hydrological model the fens in adjacent Lompolojänkkä were in general ground-water recipients (Table 3) and this is likely scenario for Lompolovuoma as well. Another thing we were considering was the fact that on the west side of the Lompolovuoma basin there is a steep-sloped fjell, which could contribute to the fact that some margins stayed wet. However, there is still a high number of the "bog-types" also on this side, and we decided to discard this explanation.

Unfortunately, no field work was conducted on this side of the Aapa mire. However, thank you for mentioning this. For further studies on the subject, these measurements would be a very good addition to data to be collected.

**Line 446: You state that the lateral spread of mire was not slowed down during the last 2000 years. Do you have information from the literature on what was the spread rate before (in an older period than the last 2000 years)?**

No data on lateral expansion from the Lompolovuoma site before our study period is available. The text in line 446 is written as a general note to reveal the contradiction between the older statements that the mire expansion has slowed down or ceased during the last 2000 years in Fennoscandia and findings

from this study that reveal that margins only initiated ca. 2200 years ago and has expanded since. This, in our opinion, is proof that mire expansion has not ceased.

**Line 462-463: You consider the slope 0.5° and higher as a steep slope not suitable for peat accumulation. I am not sure if I understand it well, but it seems to me that 0.5° is a very low value. I think the mires can origin also in steeper slopes in Central Europe and elsewhere in the mountains. Please explain in more detail the mechanism of why the peat cannot accumulate under such conditions.**

Thank you for pointing this out. In studies cited, the 0.5° threshold has been noted to slow down the peat formation process in boreal peatlands as not as much water is retained in these slopes than flatter areas. Thus, formation of anoxic conditions due to standing water is restricted, but not completely absent. However, in conditions where there are large amounts of water available, such as on mountainous regions (for example in Svalbard), maritime conditions (e.g., blanket bogs in British Isles), and areas with springs, the peat formation can occur in very steep hills as well.

To make more clear that this is due the effect of anoxic conditions (or lack of them), we will edit the text on line 462-463 accordingly:

In Lompolovuoma, the peat initiation occurred on slopes on average exceeding 0.5°, a threshold known to restrict waterlogging and thus slow down peat formation in more continental regions where availability of water is not excessive (Almquist-Jacobson & Foster, 1995; Loisel et al. 2013, Zhao et al. 2014).

**Lines 485-486: You state here that no charcoal was found in basal layers, but looking at the table with dating, I see that in the basal layer of T1Sp (40-41 cm), the charred wood was dated as well as in the case of T3Sm (19-20 cm). It indicates that some fires maybe influenced the mire margin development in some places in the past.**

Thank you for noticing this. Although there was some evidence of fire on basal layer of T3Sm based on the dated charred wood, based on the closer examination of the sample as presented in the supplementary figure S3c no charcoal was detected in this analysis. An in-situ forest fire would show high numbers of charcoal on the basal layer as seen for example in study by Juselius-Rajamäki et al. (2023) where tens or hundreds of charcoal pieces were found in basal layers associated lateral expansion influenced by forest fires. Thus, we concluded that the forest fires were not likely affecting the lateral expansion here as they are not showing in numbers sufficient enough.

To prevent any misunderstanding by readers, we add the following text to line 486.

However, in our basal layers only a single charred wood piece used for dating was found, while microscopic analysis of the basal layers did not reveal any charcoal (Supplementary figure 3a, 3b, 3c). Thus, forest fires did not likely play an important role in the peat initiation in question.

**Line 497-498 (524-525): Here is the indication of C-E stage wetness contradictory: *Pleurozium* and *Cenococcum* sclerotia speak for dry conditions and Cyperaceae for wet – but maybe *Eriophorum vaginatum* can also grow under drier or more fluctuating conditions. Maybe the conditions were really rather dry, and therefore, sphagna did not colonise it – there are many sphagnum species which are able to grow in the water (S. riparium, majus, fallax etc.) but it depends on the alkalinity and mineral richness. Do you know the parameters of the groundwater?**

Thank you for this comment. Considering the fluctuating hydrological conditions, several factors support this idea.

- Firstly, the period of the C-E stage spanned from ca. 20 to 1720 CE in the margin. During this period, both humid (The Little Ice Age) and dryer climate (Medieval Climate Anomoly) phases occurred. As no single dry climate stage persisted, at least climate conditions don't support only dry conditions (Luoto & Nevalainen 2015).
- In Aapa mires (patterned fens), important source of the water to the mires comes from snow melt. During the spring Aapa mires become largely inundated, also supporting that at least some part of the year these locations have received large amounts of water.
- Although *Eriophorum vaginatum* can indeed endure periods of droughts, for example. Wein (1973) and Tallis (1964) note that longer periods of water level drawdown led to death of cottongrass vegetation while high water tables seem be favourable for E. vaginatum (Lavoie et al. 2005).
- In addition to *Eriophorum vaginatum*, *Carex limosa* was identified in the C-E stage vegetation as shown in supplementary figure 3b and 3c. Carex limosa is generally found in areas at least partly flooded, again suggesting fluctuating hydrological regime (Visser et al. 2000).

However, as stated the absence of Sphagnum mosses raises the question of the hydrological state of these areas during the initiation. We concur that the conditions could have been periodically (or even over several years) dry (which would also explain highly decomposed peats) but nevertheless claim that the hydrological conditions were fluctuating and only after the autogenic changes wrought by Cyperaceous vegetation were the conditions suitable for Sphagnum mosses. However, as you mentioned in the other comment (for Table 1), there is also possibility that some hiatuses have occurred at certain time periods.

Unfortunately, no parameters for groundwater alkalinity were measured and the only indication of the ground water alkalinity and mineral richness (or lack of it) is the past vegetation remains in peat. The snowmelt generally provides mineral rich waters to Aapa mires, which could have prevented the Sphagnum colonization initially. However, as no rich fen associated brown mosses were found in past vegetation the alkalinity was not likely the case here. Currently the pH is ca. 4 in studied mire basin of Lompolovuoma while it is ca. 4-5 in the adjacent mire basin of Lompolojänkkä.

We have included following addition to text line 537:

*, for example Carex limosa (Supplementary figure 3b, 3c)... (Visser et al, 2000).*

Lavoie et al. 2005. The dynamics of a cotton-grass (Eriophorum vaginatum L) cover expansion in a vacuum-mined peatland, southern Quebec, Canada. Wetlands 25(1): 64-75.

Luoto, T. & Nevalainen, L. 2015. Late Holocene precipitation and temperature changes in Northern Europe linked with North Atlantic forcing. Climate research 66: 37-48.

Tallis, J. H. 1964. Studies on southern Pennine peats III: The behaviour of Sphagnum. Journal of Ecology 52(2): 345-353.

Visser et al. 2000. Flooding tolerance of Carex species in relation to field distribution and aerenchyma formation. New Phytol. 148: 93-103.

Wein, R. W. 1973. Biological flora of the British Isles: Eriophorum vaginatum L. Journal of Ecology 61(2): 601-615.

Line 515: Wouldn´t be better to plot *Eriophorum vaginatum* separately from other Cyperaceae in the diagrams? Did you identify tissues or spindles?

Thank you for this suggestion. When the decomposition of the peat samples allowed *Eriophorum vaginatum* was detected individually from Cyperaceous vegetation. In these cases, the identification was conducted either from tissue or from spindles. These results are shown in supplementary figures 3a-c.

*E. vaginatum* is plotted individually in the traditional vegetation diagrams in supplementary data, which can be accessed easily. Thus, we did not find necessary to plot in vegetation type diagrams.

Lines 541: What about the changes in the catchment area? Drainage and deforestation? Couldn´t they also influence the shift to drier bog vegetation? You write that it is good for the climate colling in the next chapter (lines 583-584), but I am afraid that this process will continue, and later, the decomposition will start and maybe even prevail. Maybe you should also discuss this possibility.

Although drainage could indeed cause drying in the mire surface there is no drainage in the area providing water for the studied mire margin. There is some forest drainage on the other side of the adjacent mire basin, but this is highly unlikely to affect the studied margin it is in adjacent catchments (the drained areas are on the east side of the adjacent Lompolojänkkä basin shown in figure 7).

On the other hand, deforestation would more likely make the area wetter and more nutrient-rich as the removal of trees would increase the waterflow from the uphill areas and increase transport of nutrients. Thus, it is not likely cause of fen-to-bog transition. In addition, apart from some small, fallen trees, no signs of large-scale deforestation were visible in the area providing water to the mire margins.

Thank you for suggesting discussing the drying trend on northern peatlands. This is an important topic and a large uncertainty in the future of the northern peatlands. We will add following to line 584:

"*However, drying trend detected in the European peatlands (Swindles et al. 2019) could also turn these locations to carbon sources, if sufficient moisture conditions are not retained (Zhang et al. 2020).*"

Swindles et al. 2019. Widespread drying of European peatlands in recent centuries. Nature geoscience 12: 922-928.

Zhang et al. 2020. Decreased carbon accumulation feedback driven by climate induced drying of two southern boreal bogs over recent centuries. Global Change Biology 26: 2435-2448.

**Line 608: Just an idea – couldn´t also play some role the relief below the mire? It could somehow influence the water flow and cause some margins to stay wet and others not.**

The topographic relief beneath the mire, particularly the break in slope in the interface between peat layers and the underlying sand or gravel formations (see Lowry et al., 2009), is a well-established factor driving localised groundwater exfiltration and the formation of springs. Different (drier/wetter) climatic conditions, however, also influence these exfiltration patterns and their intensity (Autio et al. 2020). The hydrological model used in this study and described in detail in Autio et al. (2023) incorporated all available information on peat depths, and thus, the simulated changes in the GW table account for the influence of peat relief on water flow paths.

Lowry, C. S., Fratta, D., & Anderson, M. P. (2009). Ground penetrating radar and spring formation in a groundwater dominated peat wetland. *Journal of Hydrology*, *373*(1–2), 68–79. https://doi.org/10.1016/j.jhydrol.2009.04.023

Autio, A., Ala-Aho, P., Ronkanen, A., Rossi, P. M., & Kløve, B. (2020). Implications of peat soil conceptualization for groundwater exfiltration in numerical modeling: A study on a hypothetical peatland hillslope. *Water Resources Research*, *56*(8), e2019WR026203.

**Minor and technical comments**:

We thank the referee for these comments and technical corrections. These will be addressed in the manuscript as per the instructions from the journal.

Line 123: *Empetrum nigrum* – do you mean s.str. or s.lat including *E. hermaphroditum*? And between *Andromeda polifolia* and *Vaccinium vitis-idaea* is erroneously in italics – it is typing error.

We did not differentiate between subspecies nigrum and hermaphroditum in this study, as both species are present in the location of the study site and their effect on the mire margin ecosystem would be most likely the same between the subspecies.  x

Line 293 and in other places in the manuscript: The terms *Cyperaceous* and *Ericaceous* vegetation. I don´t think that using the declension of the Latin names (in italics) is OK. I think you should use the term "*Cyperaceae*- and *Ericaceae*-dominated vegetation" or cyperaceous and ericaceous vegetation without italics and with small letters. It is the same as like *Sphagnum* species or sphagna (not *Sphagna*). x

Lines 295-296: Cyperaceous-Ericaceous-Sphagnum vegetation: I suggest using *Cyperaceae-Ericaceae-Sphagnum* instead. x

Line 297: *Sphagnum* sect. Acutifolia – also Acutifolia should be in italics. x

Line 343: *Sphagna* should not be in italics. x

Line 349: Word *sclerotia* should not be in italics. x

Line 491: "*Sphagna* is frequently found" – it should be "sphagna are …" x

Line 511: Sphagnum mosses – Sphagnum should be in italics. x

Line 571: The heading of the paragraph should be in bold.x

**Citation**: https://doi.org/10.5194/egusphere-2024-2102-RC1

**REFEREE 2**

**OVERVIEW**

**This study uses palaeoecological analysis of shallow peat cores along transects between the edges and the interior of a peatland in Finland to assess rates and mechanisms of lateral peat expansion at the site during the last ~3,000 years. The authors combine the peat cores with remote sensing analysis to identify different vegetation zones, and a hydrological model to study water flowpaths and sources. The main findings are that the peatland has indeed expanded and appears to be continuing to do so; and that the plant communities at the expanding edge differ between the three transects. The study is quite site specific, although the authors suggest that the heterogeneity in plant communities between different parts of the expanding edge gives the study some wider significance. The findings are likely to be of interest to some readers of Biogeosciences. I do not have any substantive concerns about the rigour or the presentation of**

**the research, but below I make some minor suggestions in the spirit of trying to help the authors improve their work.**

We thank you for the suggestions and corrections to improve our article. We have answered all of the suggestions given and have responded to them below. The referee comments have been colored violet and bolded, while our responses have been colored green. Any additions to the manuscript have been underlined, as per journal instructions no changes in the article should be made before the end of the review process.

**MINOR SUGGESTIONS**

**The title of the article doesn't make much sense, particularly the prefix before the colon, which claims that "Mire edge is not a marginal thing". Can this somewhat obscure prefix be deleted without removing important meaning?**

The prefix can be removed. This requires changing the word order in the title. We suggest the following new title: External and internal factors behind the formation, vegetation succession, and carbon balance of subarctic fen margin.

**76-79: This seems to contradict the findings of Evans et al. (2021), which showed that the reduction in CH4 emissions from deepening water tables is more than outweighed by increased CO2 emissions.**

Thank you for bringing this interesting article to our attention. By studying Evans et al. (2021) there appears to be an optimal WTD ca. 2.5 – 7.5 cm where the GHG balance is negative (fig 1d). When WTD is closer than ca. 2.5 cm to the surface (or above) methane becomes the controlling factor, while when the WTD is below 7.5 cm the CO2 emissions control the GHG balance. These calculations are based on 100-year global warming potential. Based on these results, conclusions are that the contradiction arises in scenario where the water table is deep enough to alter the CO2 emissions (e.g. 7.5 cm or more in this dataset) and turn GWP to positive. Based on these conclusions we will change the text in lines 76-79 to following with added text underlined:

However, in areas where the acrotelm i.e., the oxic and biologically active layer of the peat, is thick most of the methane is oxidated to carbon dioxide (Lai, 2009). Thus, in the peatland margins where dry bog-type vegetation communities dominate, the climate forcing is most likely negative, i,e, cooling impact on climate, due to the continuous uptake of CO 2 and low CH 4 emissions. However, if the water table depth becomes too deep, accelerated decomposition can turn these locations to carbon sources due to increased CO2 emissions that negate the decrease in CH4 emissions (Evans et al. 2021).

**220-234: This section is hard to follow. I realise that the reader is referred to another publication for the full method, but some more details here would be appropriate and help the current article to read as a standalone piece. What is the source of the remote sensing data? For what year(s) were they captured, and at what resolution? How do the field and remote sensing data mentioned in the paragraph fit together?**

We will modify the text to more detailed version in method sections of our paper as follows:

2.6 Current vegetation community coverage analysis

We used field and remote sensing-based land cover type data presented in more detail in (Räsänen et al., 2021) to estimate the proportion of vegetation communities in the peatland margins. Land cover classification was based on field verification data collected in summer 2019, and multisource remote sensed data. Classification was conducted in two steps: first 4 channel 0.5 m pixel sized aerial image

from summer 2018 was segmented, and then for these segments (mean size 50 m2) values were calculated from several Lidar, Planetscope and Sentinel images from years 2018 and 2019, and these were classified using random forest classification. Final land cover product had 16 classes, and the overall classification accuracy was 76%. Here, we use simplified version of this classification based on ombrotrophic – minerotrophic gradient to describe habitat conditions and related vegetation community. In addition, tree-covered fens were separated from open fens. Applied vegetation communities are: "bog"-type (referring to dry conditions), "fen"-type (referring to wet conditions), and tree-covered fens (referring to forested peatland) and these enable comparison with the remote sensing data. These were combined from the land cover type classes with similar ecological characteristics: dwarf shrub pine bogs and dwarf shrub bogs as the bogs, tall sedge fens and flarks as the fens and paludified spruce, birch, and mixed forests as the tree-covered fens. We delineated our study basin Lompolovuoma and adjacent Lompolojänkkä basin based on the land cover dataset in ArcGis Pro ver. 3.1.0 (ESRI, 2023) and calculated the proportion of each land cover type for the whole peatland area and for the peatland margins. For the peatland margins, we chose a 25-meter distance from the peatland forest border to represent the marginal peatland area. This distance prevented any overlap of the marginal areas even in the narrowest parts of the peatland and allowed non-biased analysis of the marginal peatland types irrelevant to the topography or vegetation on site.

240: what is the spatial resolution?

HydroGeoSphere model use a 3-dimensional 17-layer triangular prism grid for the subsurface flow domain with five top subsurface layers, each 10 cm thick. The top of the 3D grid was defined using a LIDAR derived 2 m × 2 m digital elevation model. All details are described in Autio et al. 2023 (https://doi.org/10.1016/j.jhydrol.2023.130342) that has been mentioned and cited in manuscript and methods section.

**446: Are there any other, more up-to-date, references for this traditional viewpoint?**

Decreasing peatland formation during the last 2000 years has been reported for example in Ruppel et al. (2013) and Korhola et al. (2010). However, these studies acknowledge the lack of data from the thinner parts of the peatlands as a more likely reason for the decrease in peatland expansion. To our knowledge more modern studies have not presented the actual cessation of peatland formation.

**575: In addition to Young et al. (2019), have a look at the follow-up article by Young et al. (2021), in which these ideas have been developed more fully, and which explains in more detail the problem to which you refer. Some consideration of the consequences for your study of this problem with apparent rates of peat/carbon accumulation would also be in order.**

We thank you for bringing this article to our attention. Considering the points raised by Young et al. (2021), discussing peat accumulation rates without any modelling is likely to give wrong impression of the early development rate of the peat layers. Similarly, using comparatives to describe peat thicknesses between different vegetation stages does not make sense. As the main topic of this article is not to assess vertical accumulation of peats and carbon accumulation, but rather horizontal development and vegetation succession, we have decided to remove the parts concentrating on the peat accumulation, e.g. lines 368 – 371 and lines 574-576.

The first paragraph in 4.3 The implications for carbon balance will be rewritten as follows:

It has been shown that in fen conditions climate forcing from peatland complex can remain positive (e.g., climate warming effect) for most of the development history due to high methane emissions and only after continuous carbon uptake and expansion of bog vegetation the climate forcing turns negative

(Korhola et al., 1996; Mathijssen et al., 2017, 2022). For example, in adjacent Lompolojänkkä basin the modelled climate warming effect persisted up to 2000 years (Mathijssen et al. 2014). Thus, it is likely that during the initial minerotrophic C-E stage lasting between 150 and 1250 years, the mire margin had a climate warming effect. Afterwards, a shift to decay-resistant Sphagnum vegetation and lower water table leading to reduced methane emissions would likely have the same effect. Decrease of the Cyperaceous vegetation especially during the last ca. 100 years would have reduced the methane emissions even further (Bubier et al., 1993; Ward et al., 2013). Although our study did not include carbon balance calculations, the shift towards bog community on the studied margins suggest that under current conditions, the margins would likely proceed to have a climate cooling effect.

**TYPOGRAPHICAL SUGESTIONS**

Thank you for these corrections. As instructed by the journal, these corrections will be done for the finalized manuscript.

35: After peatland initiation

48: areas adjacent

50: an adjacent peatland / adjacent peatlands

84, 88: deserved seems the wrong word choice here. How about attracted?

102: a fully integrated

126: we used three additional short profiles

221-222: where the methodology is described in the detail

223: we used a simplified classification

238-239: detected in our vegetation coverage analysis

300, 302 and throughout: ericaceous does not need to be italicised, as Ericaceae is a family, not a genus.

447: findings

462: In Lompolovuoma, peat initiation occurred

**References cited in this review**

Evans CD, et al. (2021) Overriding water table control on managed peatland greenhouse gas emissions. *Nature*, **593**, 548-552.

Young DM, et al. (2019) Misinterpreting carbon accumulation rates in records from near-surface peat. *Scientific Reports*, **9**, article 17939.

Young DM, et al. (2021) A cautionary tale about using the apparent carbon accumulation rate (aCAR) obtained from peat cores. *Scientific Reports*, **11**, article 9547.

**Citation**: https://doi.org/10.5194/egusphere-2024-2102-RC2

---

## Author Response (AR2)

Response to the editor's comments

We thank the editor for these comments. We have made the following changes to our manuscript:

1. We have unified the description of the age to cal. BP across the manuscript.
2. We have clarified the aims of the study in the last paragraph of the introduction section as well as removed to specifics of the methods. The methods are clearly described in the methods section and thus no changes or additions were made to the methods section.
3. We have renamed the figures in the supplementary material according to the guidelines, e.g. Fig. S1, Fig. S2. and so forth.
4. We have updated the reference list according to your standards.
5. In addition, some minor spelling mistakes were corrected and some minor upgrades to follow the house standards were done.

These comments (apart from the supplementary figures correction) is marked in the author's track changes version of the manuscript.

Please note that this document is an addition to the author's response to the reviewers comments submitted earlier and accepted by the editor.